# Trade-off between Payoff and Model Rewards in Shapley-Fair Collaborative Machine Learning

**Quoc Phong Nguyen, Bryan Kian Hsiang Low[§], and Patrick Jaillet[†]**
Institute of Data Science, National University of Singapore, Republic of Singapore
Dept. of Computer Science, National University of Singapore, Republic of Singapore[§]
Dept. of Electrical Engineering and Computer Science, MIT, USA[†]
qphongmp@gmail.com, lowkh@comp.nus.edu.sg[§], jaillet@mit.edu[†]

## Abstract

This paper investigates the problem of fairly trading off between payoff and model rewards in collaborative *machine learning* (ML) where parties aggregate their datasets together to obtain improved ML models over that of each party. Supposing parties can afford the optimal model trained on the aggregated dataset, we propose an allocation scheme that distributes the payoff fairly. Notably, the same scheme can be derived from two different approaches based on (a) desirable properties of the parties' payoffs or (b) that of the underlying payoff flows from one party to another. While the former is conceptually simpler, the latter can be used to handle the practical constraint on the budgets of parties. In particular, we propose desirable properties for achieving a fair adjustment of the payoff flows that can trade off between the model reward's performance and the payoff reward. We empirically demonstrate that our proposed scheme is a sensible solution in several scenarios of collaborative ML with different budget constraints.

## 1 Introduction

While *machine learning* (ML) has proven its usefulness in a wide range of applications, a significant amount of diverse training data is often required to achieve desirable performance. Although this may not be an issue for very large corporations, it can hardly be met by other businesses. In practice, the dataset coming from an individual data owner (e.g., a company), called a *party*, is often limited in size (e.g., due to the size of the customer base) and diversity (e.g., due to the constraints in the geographical location and/or the customer demographics). Hence, there is a strong urge for parties to obtain high-quality ML models by performing *collaborative ML*: training models on their aggregated dataset which is often of much larger quantity and higher diversity than those of individual parties.

However, the goal of obtaining improved models may not outweigh the concern of *fairness* if all parties receive the same model while their contributions differ (due to the difference in the size and quality of different parties' datasets). For instance, a party with a larger contribution feels "under-rewarded" if all parties receive models of *equal performances* without any other *compensation/rewards for the difference in their contributions*. Hence, a "fair" reward allocation scheme is desirable to give all parties enough incentives to join the collaboration. Let us classify existing works on fair reward allocation schemes into 2 groups depending on whether there exists an external source of reward to be distributed among parties. This source of reward can be thought of as an *external buyer* of the model who is interested in the model but does not contribute data, so he/she is willing to pay a reward to parties in the collaboration to obtain the model.

As an example, a company distributes monetary payoffs to its users for contributing user data to train a model. Then, one can view users as parties who collaborate to train the model and the company as an external buyer of the model. In this case, one can compensate for the difference in users' contributions

by fairly distributing the total monetary payoff from the company to users according to the value of each user's data to the model. This value can be measured by a data valuation method [15], e.g., the Shapley value [5] and its variants to incorporate the underlying data distribution [4], replication-robustness [6, 20], the federated learning setting [19], the training-free setting [18], the online collaborative learning setting [2], and to speed up the computation [3, 5, 12, 22].

In contrast, we focus on the other group of works when the obtained model from the collaboration of several parties is not sold to any external buyer (e.g., due to privacy or the fear of losing the competitive advantage), i.e., there is not any external source of reward to be distributed among parties. In the literature, fairness is maintained by (a) requiring parties to pay a participation fee to be distributed back to all parties [13] or (b) ensuring that a party with less valuable data receives a model with lower performance than those with more valuable data [16]. In brief, parties receive different *monetary payoffs* (produced by the participation fee) in the former and different models (called *model rewards*) in the latter. However, there are 2 main drawbacks: the trading-off between monetary payoffs vs. model rewards and the uniqueness of the allocation scheme. **First**, the former approach [13] prevents parties with an insufficient monetary budget from joining the collaboration regardless of the usefulness of their datasets to other parties. The latter approach [16] prevents parties with low-quality datasets from obtaining a good model regardless of their monetary expense capability. Both cases are undesirable. **Second**, the uniqueness of the allocation schemes in the works of [13, 16] is not discussed. Hence, it raises the question of whether there are other allocation schemes that satisfy the same set of fairness properties in [13, 16]. If such schemes exist, there can be undesirable disagreement among parties in choosing the allocation scheme to use. On the contrary, if an allocation scheme is uniquely defined from a set of fairness properties (e.g., the celebrated Shapley value [14]), parties which agree on these properties are certain to adopt the (unique) allocation scheme.

In this work, we resolve the above 2 problems by proposing fairness properties that define not only a unique allocation scheme of the monetary payoff but also a unique way of trading off between payoff and model rewards. It is noted that as both payoff and model rewards are allowed, our problem subsumes the two problems in [13, 16]. Nevertheless, our solution is not a generalization of existing solutions [13, 16] because even in the special cases of our problem that are equivalent to those in [13, 16], our solution is different from that of [13, 16]. In particular, our solution satisfies the linearity property while the works of [13, 16] do not. The outline of our paper is as follows.

In Sec. 3, we consider parties without budget constraints, which implies that all parties can afford the highest-quality model from the collaboration. Thus, the scheme is reduced to only allocating monetary payoffs to parties. In this case, we propose 4 desirable properties which uniquely determine the fair payoff for each party (in Sec. 3.1). The resulting payoffs *fairly compensate for the difference between the contributions (measured by the Shapley value) of parties to the models they receive*, which we call *Shapley fairness*. While these payoffs suffice to characterize a payoff allocation scheme, it does not give us insights into the underlying payoff flows transferred from one party to another. The latter is crucial for us to unravel the unexplored question of trading off between payoff and model rewards when parties have budget constraints. Therefore, in Sec. 3.2, we propose 4 desirable properties for fair payoff flows by drawing inspiration from the Shapley value [14]. These properties uniquely determine a new solution called *the conditional Shapley value* which specifies the payoff flows from one party to another. Interestingly, we arrive at the same set of payoffs as those in Sec. 3.1. Hence, the fairness of our proposed payoff allocation scheme can be consistently explained with both the proposed properties of the payoffs (Sec. 3.1) and those of the underlying payoff flows transferred from one party to another (Sec. 3.2).

In Sec. 4, we consider parties with budget constraints, i.e., their expense cannot exceed their budgets. Consequently, there may exist parties that cannot afford the highest-quality model if their budgets are below the required amount of payment. We propose to trade off between payoff and model rewards such that parties can sacrifice the model reward, i.e., obtaining a worse model, to meet the budget. In particular, we construct a set of desirable properties that uniquely define adjusted value functions from which the resulting payoff rewards both meet the budgets and satisfy fairness properties.

In Sec. 5, we empirically demonstrate sensible characteristics of our proposed allocation scheme in several ML problems. For example, even if parties do not have any budget, they are still motivated to join the collaboration because they are able to obtain improved models as long as they have valuable datasets. Furthermore, an increase in the budget of a party not only benefits itself but also benefits other parties through improved models and/or higher monetary payoffs.

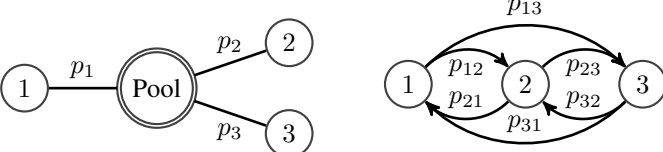

(a) Payoff reward representation.    (b) Payoff flow representation.

Figure 1: Diagrams of the representations of a payoff allocation scheme.

## 2 Background

Let us consider a set of $n$ parties, denoted as $\mathcal{N} \triangleq \{1, 2, 3, \ldots, n\}$, collaborating with each other by aggregating their datasets to train ML models. Any subset of $\mathcal{C} \subset \mathcal{N}$ is a *coalition* while $\mathcal{N}$ is the *grand coalition*. To ease the notation clutter, we overload the notation $i$ to denote both party $i$ and the singleton set $\{i\}$. The aggregated training dataset of a coalition $\mathcal{C}$ is denoted as $\mathcal{D}_{\mathcal{C}} \triangleq \cup_{i \in \mathcal{C}} \mathcal{D}_i$ where $\mathcal{D}_i$ denotes the dataset of party $i$. The performance of a model trained on $\mathcal{D}_{\mathcal{C}}$ is denoted as $s(\mathcal{C})$. We assume the *monotonicity property* of $s$ with respect to (w.r.t.) $\mathcal{C}$: $\forall \mathcal{C} \subset \mathcal{C}' \subset \mathcal{N}$, $s(\mathcal{C}') \geq s(\mathcal{C})$; $s(\emptyset) = 0$, and $s(\mathcal{N}) > 0$. In the experiments (in Sec. 5), the model performance is measured by the prediction accuracy of models on a validation set which is common across different parties. It reflects the assumption that parties solve the same ML problem. Our approach also works with other performance metrics such as the information gain in the work of [16].

The *value function*, denoted as $v$, is a mapping from a coalition $\mathcal{C} \in \mathcal{N}$ to its *value* $v(\mathcal{C})$ such that $v(\emptyset) = 0$. In this work, $v(\mathcal{C})$ is considered as the monetary payoff. Adopting the assumption from the work of [13], we define it using an exchange rate $\nu > 0$: $v(\mathcal{C}) \triangleq \nu s(\mathcal{C})$. From the properties of $s$, it follows that $v$ is monotonic, $v(\emptyset) = 0$, and $v(\mathcal{N}) > 0$. More importantly, *a model with the performance $s(\mathcal{C})$ can be exchanged fairly with a payoff $v(\mathcal{C}) \triangleq \nu s(\mathcal{C})$ and vice versa.*

Given a value function $v$, the Shapley value is the solution to distributing the payoff $v(\mathcal{N})$ to parties in $\mathcal{N}$ that satisfies 4 desirable properties: efficiency, symmetry, linearity, and dummy party (see App. A). Hence, it is the foundation of many data valuation methods [5, 4, 7, 8, 21]. The Shapley value of a party $i$ is defined as $\varphi_i(v) \triangleq \frac{1}{n} \sum_{\mathcal{C} \in \mathcal{N} \setminus i} \binom{n-1}{|\mathcal{C}|}^{-1} \Delta_v(i|\mathcal{C})$ where $\Delta_v(i|\mathcal{C}) \triangleq v(\mathcal{C} \cup i) - v(\mathcal{C})$ is the marginal contribution of party $i$ to coalition $\mathcal{C}$. It is noted that to use the Shapley value, there should exist a payoff $v(\mathcal{N})$ to be distributed to all parties in $\mathcal{N}$, i.e., there exists an external source of reward (see the classification of existing works in Sec. 1). In contrast, we assume that such an external source of reward does not exist in this work, which renders the direct use of the Shapley value challenging. Thus, we propose a new perspective of distributing a payoff of value $0$ to all parties in $\mathcal{N}$ (by allowing parties to receive negative payoff rewards) such that these payoff rewards *compensate for the difference in the contributions among parties.*

In this work, an *allocation scheme* specifies both payoff and model rewards that each party receives. We do not distinguish between two models having the same performance. Thus, for a party $i \in \mathcal{N}$, the scheme specifies the value (equivalently, the performance) of the model and the payoff value it receives. To receive a negative payoff is to make a payment.

We first consider the case where there are not any budget constraints (i.e., unlimited budgets), so all parties can afford the best model with the performance of $s(\mathcal{N})$ (i.e., the optimal model trained on $\mathcal{D}_{\mathcal{N}}$) while they may receive different payoff rewards to account for the difference in their contributions in Sec. 3. Then, in Sec. 4, we consider the case that parties have budget constraints such that they may not afford the model of performance $s(\mathcal{N})$. In this case, not only do parties receive payoffs constrained by the budgets, but they also receive different model rewards to maintain fairness.

## 3 Parties without Budget Constraints

In this section, parties have no budget constraints, so they can expend any amount of payoff to receive the optimal model trained on $\mathcal{D}_{\mathcal{N}}$ (i.e., with the performance of $s(\mathcal{N})$). Hence, the problem is reduced

to an allocation scheme of only the payoff reward. In particular, different parties receive different (and possibly negative) payoff rewards to compensate for the difference in their contributions.

**Payoff reward vector.** Given a value function $v$, let $p_i(v)$ denote the payoff reward that party $i$ receives. Then, a payoff allocation scheme specifies a payoff reward vector $\mathbf{p}(v) \triangleq (p_i(v))_{i=1}^n$. While party $i$ is rewarded if $p_i(v) > 0$, it makes a payment of value $-p_i(v)$ if $p_i(v) < 0$. In practice, it can be implemented as a 2-step process: first, parties with negative payoffs make payments to an intermediate pool; second, parties with positive payoffs take their payoffs from the pool (Fig. 1a).

**Payoff flow.** One may consider defining a payoff allocation scheme by the payoff transferred from one party to another which we call *payoff flows* (Fig. 1b). However, there can be different ways of transferring payoffs among parties which result in the same payoff reward vector. For example, a payoff reward vector remains unchanged if we increase each payoff flow from party $i$ to party $\tau(i)$ by the same amount where $\tau(i) = i + 1$ for all $i < n$ and $\tau(n) = 1$ (i.e., increasing the payoff along a closed loop). Furthermore, regardless of how payoffs are transferred among parties, what matters, in the end, is the payoff reward each party receives which is determined by the payoff reward vector $\mathbf{p}(v)$. Therefore, in the next section, we propose 4 desirable properties that uniquely determine a fair payoff reward vector (hence, a payoff allocation scheme).

## 3.1 Fair Payoff Reward Vector

We propose the following 4 desirable properties for the fairness of a payoff reward vector $\mathbf{p}(v)$. While the symmetry and linearity properties are taken from that of the Shapley value, the efficiency and dummy party properties are modified to fit our problem setting where there is not any external source of reward to be distributed among parties.

**Symmetry.** If $v(\mathcal{C} \cup i) = v(\mathcal{C} \cup j)$ for all $\mathcal{C} \subset \mathcal{N} \setminus \{i, j\}$, then $p_i(v) = p_j(v)$. It implies parties with equal marginal contributions to any coalitions have the same payoff.

**Linearity.** Let us consider the value function as a vector in a real vector space of dimension $2^n$ (i.e., the number of subsets of $\mathcal{N}$) which is equipped with the usual addition and scalar multiplication. Then, given 2 value functions $v$ and $w$,

$$p_i(v + w) = p_i(v) + p_i(w) , \quad p_i(\alpha v) = \alpha p_i(v) , \quad \forall i \in \mathcal{N} , \forall \alpha \in \mathbb{R} .$$

**Balance.** Recall that there is not any external source that pumps additional payoffs into $\mathcal{N}$ (or takes payoffs from $\mathcal{N}$). Therefore, the *balance* property states that $\sum_{i \in \mathcal{N}} p_i(v) = 0$.

**Dummy party.** Party $i \in \mathcal{N}$ is a dummy party if $v(\mathcal{C} \cup i) = v(\mathcal{C})$ for all $\mathcal{C} \subset \mathcal{N} \setminus i$. It means that a dummy party can be considered as an external buyer that does not contribute any data and is only interested in buying the model. Therefore, we expect a dummy party to pay the whole value of the model it receives from the collaboration, i.e., $v(\mathcal{N})$ (since parties have no budget constraints). In other words, if party $i$ is a dummy party, then it makes a payment of value $v(\mathcal{N})$, i.e., $p_i(v) = -v(\mathcal{N})$.

It is noted that we do not sacrifice the uniqueness of the Shapley value by modifying its properties. The following lemma shows that these newly-proposed properties still characterize a unique payoff reward vector, which is proven in App. B.

**Lemma 3.1.** *There exists a unique vector $(p_i(v))_{i=1}^n$ of payoff rewards that satisfies the balance, the symmetry, the linearity, and the dummy party properties:*

$$\forall i \in \mathcal{N}, \ p_i(v) = n\varphi_i(v) - v(\mathcal{N}) . \tag{1}$$

**Corollary 3.2.** *If we allocate payoff rewards to parties in $\mathcal{N}$ according to (1), then a party has a negative payoff reward (i.e., making a payment) if and only if its Shapley value is less than the average of the Shapley values of all parties, i.e., $v(\mathcal{N})/n$. This implies that a party receives a negative payoff if its contribution is less than the average of the contributions of all parties.*

We also study another property, namely *replication-robustness* [1, 6], in App. D where we show that the proposed payoff allocation scheme is replication-robust for parties with negative payoffs.

**Corollary 3.3.** *We observe that $p_i(v)$ is a linear function of $\varphi_i(v)$:*

$$\forall \{i, j\} \subset \mathcal{N}, \ p_i(v) - p_j(v) = n(\varphi_i(v) - \varphi_j(v)) . \tag{2}$$

While the Shapley values are often scaled to ensure all of the rewards are distributed (i.e., being efficient) [13, 16], the scaling transformation violates not only the linearity property but also the above balance property: if all parties have positive Shapley values, any payoff rewards proportional to the Shapley values (by scaling) are either all positive or all negative. Unlike the scaling transformation, the linear transformation in Corollary 3.3 allows a party with a positive Shapley value to have a negative payoff reward. Furthermore, it realizes our intuition of fairness: *the payoff rewards compensate the difference in the contributions (measured by the Shapley value) between parties*, which we call *Shapley fairness*. In particular, a larger difference in the contributions of two parties leads to a larger difference in their payoff rewards, and parties with higher contributions receive higher payoff rewards.

*Remark* 3.4. While the Shapley value represents the contribution of a party to the optimal model trained on $\mathcal{D}_{\mathcal{N}}$, there should be $n$ such models distributed to $n$ parties. Thus, the contribution of a party to $n$ models should be $n\varphi_i(v)$, which explains the factor $n$ in (2). The fact that data is *freely replicable* (e.g., to $n$ parties) is a special property of data that differentiates itself from the usual commodity studied in cooperative game theory [1, 16].

*Remark* 3.5. From (1), one interpretation of the allocation of $p_i(v)$ is as follows: first, every party deposits $v(\mathcal{N})$ to a common pool as if they were dummy parties (i.e., they have not made any contribution); second, parties contribute their datasets to form the aggregated dataset $\mathcal{D}_{\mathcal{N}}$ from which the optimal model is trained; third, parties receive the optimal model and also parts of the pool of value $nv(\mathcal{N})$ (deposited by all parties in the first step) according to their contributions to the optimal model measured by the Shapley values, i.e., $n\varphi_i(v)$.

*Remark* 3.6. We note that the above modification of the efficiency and dummy party properties can be generalized while maintaining the uniqueness of the payoff reward vector. This can be of indepedent interest to readers. In particular, there exists a unique payoff reward vector that satisfies the symmetry property, the linearity property, the generalized efficiency property: $\sum_{i\in\mathcal{N}} p_i(v) = \beta$, and the generalized dummy party property: $p_{i'}(v) = (-\alpha v(\mathcal{N}) + \beta)/n$ for any dummy party $i'$ and scalars $\alpha, \beta$. It is specified by $p_i(v) = \alpha\varphi_i(v) + (\beta - \alpha v(\mathcal{N}))/n$. For example, $\mathbf{p}(v)$ is the Shapley value for $\alpha = 1$ and $\beta = v(\mathcal{N})$, while $\mathbf{p}(v)$ is our proposed payoff reward vector in Lemma 3.1 for $\alpha = n$ and $\beta = 0$. When $\alpha = 0$, every party receives the same payoff of value $\beta/n$.

Although a payoff allocation scheme is fully determined by the payoff reward vector $\mathbf{p}(v)$, it remains an open question how one fairly adjusts the payoff reward under budget constraints. Thus, the next section describes a particular way to specify payoff flows from one party to another such that it gives rise to the same desirable payoff vector as the one in Lemma 3.1. The notion of payoff flow allows us to handle budget constraints as explained later in Sec. 4.

## 3.2 Fair Payoff Flows

Let us decompose the payoff reward $p_i(v) \in \mathbb{R}$ into the *total cost flow* $p_{i,-}(v) \geq 0$ (from party $i$ to others) that party $i$ needs to pay to obtain the optimal model trained on $\mathcal{D}_{\mathcal{N}}$ and the *total revenue flow* $p_{i,+}(v) \geq 0$ (from others to party $i$) that party $i$ obtains by contributing its dataset to the models other parties receive (the revenue does not deduct any cost):

$$p_i(v) = p_{i,+}(v) - p_{i,-}(v) . \tag{3}$$

Since $p_{i,-}$ is the total cost that party $i$ incurs to acquire the additional performance of the model from the datasets of other parties $\mathcal{N} \setminus i$, the total cost flow can be decomposed into

$$p_{i,-}(v) = \sum_{j\in\mathcal{N}\setminus i} p_{ij}$$

where the payoff flow $p_{ij}$ denotes the cost that party $i$ pays to party $j$ for party $j$'s contribution. Equivalently, it is the revenue that party $j$ obtains from party $i$. Hence, the total revenue flow is decomposed into

$$p_{i,+}(v) = \sum_{j\in\mathcal{N}\setminus i} p_{ji} .$$

Note that the payoff flow $p_{ij}(v)$ can be viewed as either a cost of party $i$ or a revenue of party $j$. Therefore, to construct all payoff flows between parties, we choose to only examine the costs of all parties because it is easier to argue about the total cost of a party (see the *fair cost* property below) than the total revenue of a party.

In particular, for a party $i \in \mathcal{N}$, we are interested in the (cost) payoff flows $(p_{ij}(v))_{j\in\mathcal{N}\setminus i}$. As there are many different ways of transferring payoff flows from one party to another that result in the same

payoff reward vector $\mathbf{p}(v)$, it is an ill-posed problem to reconstruct payoff flows solely from $\mathbf{p}(v)$ in Lemma 3.1. Therefore, we propose the following desirable properties of fair payoff flows that are able to not only uniquely determine $(p_{ij}(v))_{j \in \mathcal{N} \setminus i}$ (we assume $p_{ii}(v) = 0$ in our construction), but also result in the same payoff reward vector as that in Lemma 3.1.

**Fair cost.** As explained above, $p_{i,-}(v)$ is the cost that party $i$ incurs to acquire the additional performance from the datasets of other parties $\mathcal{N} \setminus i$. This additional performance is the difference between the performance $s(\mathcal{N})$ of the obtained model trained on $\mathcal{D}_{\mathcal{N}}$ and the performance $s(i)$ of the model trained on the dataset of party $i$. Equivalently, the increase in the value of the model is $\nu(s(\mathcal{N}) - s(i)) = v(\mathcal{N}) - v(i)$. Therefore, we suggest the *fair cost* property: the total cost flow $p_{i,-}(v)$ of party $i$ should be equal to the value of the additional performance obtained by party $i$:

$$p_{i,-}(v) = v(\mathcal{N}) - v(i), \text{ equivalently, } \sum_{j \in \mathcal{N} \setminus i} p_{ij}(v) = v(\mathcal{N}) - v(i) . \tag{4}$$

**Conditional symmetry.** If two parties $j, k$ in $\mathcal{N} \setminus i$ have the same marginal contribution to coalition $\mathcal{C} \cup i$ for any $\mathcal{C} \subset \mathcal{N} \setminus \{i, j, k\}$, then the cost party $i$ pays to party $j$ is the same as that party $i$ pays to party $k$, i.e., $p_{ij}(v) = p_{ik}(v)$. It also implies that the cost of party $i$ does not depend on the marginal contributions of other parties to coalitions that do not contain $i$.

**Linearity.** Let us consider the value function as a vector of $2^n$ real numbers with the usual addition and scalar multiplication operations, then for value functions $v, w$,

$$p_{ij}(v + w) = p_{ij}(v) + p_{ij}(w) , \quad p_{ij}(\alpha v) = \alpha p_{ij}(v) , \quad \forall i, j \subset \mathcal{N} , \forall \alpha \in \mathbb{R} .$$

**Conditional dummy party.** A conditional dummy party $j \in \mathcal{N} \setminus i$ given a party $i$ satisfies $v(\mathcal{C} \cup \{i, j\}) = v(\mathcal{C} \cup i)$ for all $\mathcal{C} \subset \mathcal{N} \setminus \{i, j\}$. For a conditional dummy party $j$ given party $i$, $p_{ij}(v) = 0$. It means that party $i$ does not pay any payoff to a party that does not contribute to any coalition containing $i$.

The above 4 properties uniquely determine the payoff flows according to the following lemma which is proved in App. E.

**Lemma 3.7.** *There exists a unique solution $(p_{ij}(v))_{j \in \mathcal{N} \setminus i}$ that satisfies the above 4 properties. We call this unique solution the* conditional Shapley value *given party $i$ and denote it as $(\varphi_{j|i}(v))_{j \in \mathcal{N} \setminus i}$. It is defined as follows:*

$$\varphi_{j|i}(v) = \frac{1}{n-1} \sum_{\mathcal{C} \in \mathcal{N} \setminus \{i, j\}} \binom{n-2}{|\mathcal{C}|}^{-1} \Delta_v(j | \mathcal{C} \cup i) . \tag{5}$$

Like the Shapley value, the conditional Shaple value given party $i$ only depends on the marginal contribution $\Delta_v(j | \mathcal{C} \cup i)$ for $\mathcal{C} \subset \mathcal{N} \setminus i$ in (5), so it is *translation invariant* as elaborated in the following corollary.

**Corollary 3.8.** *If $v'(\mathcal{C} \cup i) = v(\mathcal{C} \cup i) + C, \forall \mathcal{C} \subset \mathcal{N} \setminus i$ where $C \in \mathbb{R}$ is a constant, then $\varphi_{j|i}(v) = \varphi_{j|i}(v')$ for all $j \in \mathcal{N} \setminus i$.*

*Remark* 3.9. Since the formulation of the conditional Shapley value is similar to that of the Shapley value, we can apply existing approximation methods of the Shapley value [3, 5, 12] to the conditional Shapley value.

*Remark* 3.10. The payoff flow $\varphi_{j|i}(v)$ from party $i$ to party $j$ is interpreted as the amount of payoff that party $i$ needs to pay party $j$ for its contribution to the model recevied by party $i$. It is also viewed as the amount of contribution that party $j$ contributes to party $i$ through any coalitions that containing party $i$. Besides, $\varphi_{j|i}(v)$ is independent of the marginal contributions of party $j$ to any coalitions that do not contain party $i$. It can be clearly observed from the alternative formulation of the conditional Shapley value:

$$\varphi_{j|i}(v) = \frac{1}{(n-1)!} \sum_{\pi \in \Pi_i} \Delta_v(j | \mathcal{P}_\pi^j)$$

where $\pi$ denotes a permutation of $\mathcal{N}$, $\mathcal{P}_\pi^j$ denotes parties preceding $j$ in $\pi$, and $\Pi_i$ denotes the set of permutations that start with $i$.

**Decomposition of the Shapley value.** The Shapley value of a party can be decomposed into the conditional Shapley values and its value as shown in App. F:

$$n\varphi_i(v) = v(i) + \sum_{j \in \mathcal{N} \setminus i} \varphi_{i|j}(v) . \tag{6}$$

We recall in Remark 3.4 that a party contributes to $n$ models (each of which is distributed to one party in $\mathcal{N}$). Hence, the total contribution of a party is $n\varphi_i(v)$ which is the LHS of (6). On the other hand, the RHS of (6) is the sum of the contributions of party $i$ to other parties $j \in \mathcal{N} \setminus i$ measured by the conditional Shapley values $\varphi_{i|j}(v)$ and its own value $v(i)$. The latter is interpreted as the contribution of party $i$ to the model it receives from the collaboration. It is $v(i)$ because party $i$ can always obtain a model of $v(i)$ from its dataset. In brief, the contribution of party $i$ (to $n$ models for $n$ parties) measured by the Shapley value can be decomposed into the contributions of party $i$ to the models that other parties receive measured by the conditional Shapley values and that to the model party $i$ receives.

We set $p_{ii} = 0$, so by setting $p_{ij}(v) = \varphi_{j|i}(v)$ for $i \neq j$, all payoff flows are fully specified. Then, for all $i \in \mathcal{N}$, the payoff reward $p_i(v)$ is

$$p_i(v) = p_{i,+}(v) - p_{i,-}(v) = n\varphi_i(v) - v(\mathcal{N}) \tag{7}$$

where the total cost flow is $p_{i,-}(v) \triangleq \sum_{j \in \mathcal{N}} p_{ij}(v) = v(\mathcal{N}) - v(i)$ (from the fair cost property) and the total revenue flow is $p_{i,+}(v) \triangleq \sum_{j \in \mathcal{N}} p_{ji}(v) = n\varphi_i(v) - v(i)$ (from (6)). The payoff rewards specified by (7) are exactly the same as that in Lemma 3.1. Thus, they also satisfy the 4 desirable properties in Sec. 3.1: symmetry, linearity, balance, and dummy party.

*Remark* 3.11 (Omnipotent party). Furthermore, if we define an *omnipotent party* as a party $i$ that satisfies $v(\mathcal{C} \cup i) = v(i)$ for all $\mathcal{C} \subset \mathcal{N} \setminus i$. Then, party $i$ does not pay to any other parties, i.e., $p_{i,-} = 0$. It is because other parties $j \in \mathcal{N} \setminus i$ are conditional dummy parties given party $i$.

In brief, when parties have no budget constraints, we present a fair payoff allocation scheme that arises from either a set of desirable properties of the payoff rewards (Sec. 3.1) or that of the underlying payoff flows (Sec. 3.2). While the former is conceptually simpler, the latter can be utilized in the next section to address the budget constraints of parties.

# 4 Parties with Budget Constraints

In the previous section, we assume all parties have no budget constraints, so they can afford the optimal model with the performance $s(\mathcal{N})$ (trained on $\mathcal{D}_{\mathcal{N}}$). However, in practice, parties often have budget constraints. Let $b_i \geq 0$ denote the budget of party $i$. When $-p_i(v) > b_i$, party $i$ cannot afford the optimal model trained on $\mathcal{D}_{\mathcal{N}}$. To allow party $i$ to join the collaboration without sacrificing fairness, we would like to reduce the performance of the model it receives such that its corresponding payoff reward satisfies the budget constraint. Intuitively, a "tight-budget" party has to buy a "cheaper" model (i.e., a model of lower quality). This implies that different parties receive models of different performances. We call it the *heterogeneous model rewards*. As we discuss in the following paragraph, while the reward vector representation based on the Shapley value is not suitable, the payoff flows based on our proposed conditional Shapley value are able to handle this heterogeneity.

We recall in the Sec. 3 where there are not any budget constraints, all parties receive the same model of value $v(\mathcal{N})$. In contrast, from the above discussion, the heterogeneous model rewards imply that different parties may receive models of different values, e.g., party $i$ and party $j$ receive models of different values, denoted as $v_i(\mathcal{N})$ and $v_j(\mathcal{N})$, respectively. It means a party may contribute to different parties differently. As the payoff reward vector in Sec. 3.1 relies on the Shapley value that does not differentiate the contributions of a party to different parties, it is challenging to devise a principled adjustment of the payoff reward vector to meet the budget constraint. On the other hand, by investigating the payoff flows from one party to another through our proposed conditional Shapley value (which is also viewed as the contribution in Remark 3.10), we can adjust the amount of contribution from one party to different parties differently.

Let us consider party $i$ which cannot afford the optimal model of value $v(\mathcal{N})$ due to its budget, i.e., $b_i < -p_i(v)$. For party $i$ to meet its budget, we need to increase $p_i(v)$ (equivalently, decreasing $-p_i(v)$) such that the adjusted payoff reward is at least $-b_i$. From the decomposition of payoff rewards into payoff flows in Sec. 3.2, the payoff reward is $p_i(v) = p_{i,+}(v) - p_{i,-}(v)$ (7). It is not possible for party $i$ to increase its total revenue flow $p_{i,+}(v)$ since it comes from other parties. Hence, the only way for party $i$ to increase $p_i(v)$ is by decreasing its total cost flow $p_{i,-}(v)$ to other parties $\mathcal{N} \setminus i$. We recall that this total cost flow is $p_{i,-}(v) \triangleq \sum_{j \in \mathcal{N} \setminus i} p_{ij}(v) = \sum_{j \in \mathcal{N} \setminus i} \varphi_{j|i}(v)$. Hence, another interpretation is that party $i$ reduces the contributions $\varphi_{j|i}(v)$ of other parties $j \in \mathcal{N} \setminus i$ to itself (in

order to decrease the cost party $i$ needs to pay to other parties). The amount of reduction in $\varphi_{j|i}(v)$ is different for different parties $i$ since they may have different budget constraints. Therefore, for each party $i \in \mathcal{N}$, we define an *adjusted value function*, denoted $v'_i$, from which the *adjusted payoff flows* $p_{ij}(v'_i)$ are uniquely defined as the conditional Shapley values $\varphi_{j|i}(v'_i)$ (Sec. 3.2).[1] It is noted that instead of adjusting the underlying value function, existing works [13, 16, 17]) directly modify the Shapley value. However, this invalidates its fairness properties which is undesirable.

Let us denote the set of these adjusted value functions as $\overrightarrow{v}' \triangleq (v'_i)_{i=1}^n$. Then, by defining $p_{ij}(\overrightarrow{v}') \triangleq p_{ij}(v'_i)$, $p_{i,-}(\overrightarrow{v}') = p_{i,-}(v'_i) = v'_i(\mathcal{N}) - v'_i(i)$, and $p_{i,+}(\overrightarrow{v}') = \sum_j p_{ji}(v'_j)$, the 4 fairness properties in Sec. 3.2 uniquely define the payoff flow as the conditional Shapley value $p_{ij}(v'_i) = \varphi_{j|i}(v'_i)$. Let $\overrightarrow{v} \triangleq (v_i)_{i=1}^n$, $v_i(\mathcal{C}) = v_j(\mathcal{C}) = v(\mathcal{C})$ for all $i$, $j$, and $\mathcal{C} \subset \mathcal{N}$ ($v$ is the un-modified value function), then $\varphi_{j|i}(\overrightarrow{v}) = \varphi_{j|i}(v)$ is the payoff flow when all parties have no budget constraint (Sec. 3.1). Given $v$ and the budget $\boldsymbol{b} \triangleq (b_i)_{i=1}^n$, we would like to find adjusted value functions $\overrightarrow{v}' \triangleq (v'_i)_{i=1}^n$ such that the payoff flows $\varphi_{j|i}(\overrightarrow{v}') = \varphi_{j|i}(v'_i)$ meet the budget constraints, i.e., $-p_i(v'_i) \leq b_i$, $\forall i \in \mathcal{N}$. To achieve this goal, we propose the following fairness properties of the adjustment.

**Feasibility & Individual Rationality.** Party $i$ is interested in reducing its cost due to the increase in the value $v(\mathcal{N}) - v(i)$. Therefore, a rational party should receive a model with the (adjusted) value $v'_i(\mathcal{N})$ that is (**feasibility**) at most $v(\mathcal{N})$; and (**individual rationality**) at least $v(i)$ (i.e., not worse than the value $v(i)$ of the model it can obtain by itself). The same argument should apply to any coalition containing $i$:

$$\forall i \in \mathcal{N}, \forall \mathcal{C} \subset \mathcal{N} \setminus i, \quad v(\mathcal{C} \cup i) \geq v'_i(\mathcal{C} \cup i) \geq v(i) . \tag{8}$$

**Marginal contribution consistency.** From an alternative perspective, party $i$ reduces its cost by reducing the contribution of other parties to the increase in the performance of the model it receives. This can be measured by the conditional Shapley values of other parties given $i$ which consist of the marginal contributions of other parties to any coalition containing $i$. Therefore, the *marginal contribution consistency* states that the marginal contributions of a party $j \neq i$ to any coalition containing $i$ should be consistently reduced by the same factor:

$$\forall i \in \mathcal{N}, \exists \lambda_i > 0, \forall j \in \mathcal{N} \setminus i, \forall \mathcal{C} \subset \mathcal{N} \setminus \{i,j\}, \Delta_{v'_i}(j|\mathcal{C} \cup i) = \lambda_i \Delta_v(j|\mathcal{C} \cup i) . \tag{9}$$

It means that the adjustment should not unfairly bias the marginal contribution of a party to a coalition.

**Budget efficiency.** The budget efficiency encourages every party to obtain the best model that it can afford by expending as much of its budget as possible. On one hand, if party $i$ can afford the optimal model of value $v(\mathcal{N})$ (i.e., $-p_i(v) \leq b_i$), then it receives the optimal model under the adjusted payoff flows, i.e., $v'_i(\mathcal{N}) = v(\mathcal{N})$. On the other hand, if party $i$ cannot afford the optimal model of value $v(\mathcal{N})$ due to its budget (i.e., $-p_i(v) > b_i$), then the adjusted payoff is exactly $-b_i$, i.e., $p_i(v'_i) = -b_i$. It means that the budget is fully consumed.

The above properties and the budget constraints uniquely define a fair adjustment of the value function as shown in the following lemma (proven in App. G).

**Lemma 4.1.** *Let* $\boldsymbol{\lambda}^* \triangleq (\lambda_i^*)_{i=1}^n$ *be the solution to the following linear programming problem:*

$$\begin{aligned} \text{maximize} \quad & \sum_{\{i,j\} \subset \mathcal{N}} \lambda_i \varphi_{j|i}(v) \\ \text{subject to} \quad & \lambda_i \sum_{j \in \mathcal{N} \setminus i} \varphi_{j|i}(v) - \sum_{k \in \mathcal{N} \setminus i} \lambda_k \varphi_{i|k}(v) \leq b_i, \quad \forall i \in \mathcal{N} \\ & \lambda_i \in [0,1], \quad\quad\quad\quad\quad\quad\quad\quad\quad\quad\quad\quad\quad\quad\quad \forall i \in \mathcal{N} . \end{aligned} \tag{P}$$

*The feasibility & individual rationality, the marginal contribution consistency, the budget efficiency, and the budget constraints uniquely determine the following adjusted value functions:*

$$\forall i \in \mathcal{N}, \forall \mathcal{C} \subset \mathcal{N} \setminus i, v'_i(\mathcal{C} \cup i) = \lambda_i^* v(\mathcal{C} \cup i) + (1 - \lambda_i^*)v(i) . \tag{10}$$

*Then, from the fairness properties in Sec. 3.2, the payoff flow* $p_{ij}(v'_i)$ *is uniquely defined as the conditional Shapley value* $\varphi_{j|i}(v'_i)$ *which can be directly computed from* $\varphi_{j|i}(v)$:

$$\forall j \in \mathcal{N} \setminus i, p_{ij}(v'_i) = \varphi_{j|i}(v'_i) = \varphi_{j|i}(\lambda_i^* v) = \lambda_i^* \varphi_{j|i}(v) . \tag{11}$$

---

[1]As the conditional Shapley value given $i$ only depends on the value function at coalitions containing $i$, we are only concerned with defining $v'_i$ on the restricted domain $\{\mathcal{C} \cup i | \mathcal{C} \subset \mathcal{N} \setminus i\}$.

We note that the LHS of the first set of constraints in (P) are the negative values of the payoffs, i.e., $-p_i(\overrightarrow{v}') = p_{i,-}(\overrightarrow{v}') - p_{i,+}(\overrightarrow{v}')$, $\forall i \in \mathcal{N}$, so they are the budget constraints.

Given the adjusted payoff flows, the total revenue flow and the total cost flow are computed as

$$p_{i,+}(\overrightarrow{v}') = \sum_{j \in \mathcal{N} \setminus i} \lambda_j^* \varphi_{i|j}(v) \text{ and } p_{i,-}(\overrightarrow{v}') = \lambda_i^* \sum_{j \in \mathcal{N} \setminus i} \varphi_{j|i}(v), \text{ respectively.}$$

The value of the model obtained by party $i$ according to the above adjusted value function is $v_i'(\mathcal{N}) = \lambda_i^* v(\mathcal{N}) + (1 - \lambda_i^*) v(i) = v(i) + \lambda_i^* p_{i,-}(v) = v(i) + p_{i,-}(v_i')$. Hence, the value of the model obtained by party $i$ is the sum of the value $v(i)$ of model trained on party $i$'s dataset and the value of the contribution $p_{i,-}(v_i')$ of other parties to party $i$. This also demonstrates the feasibility & individual rationality property (since $\lambda_i \in [0, 1]$). To produce a model with a pre-specified performance (e.g., the validation accuracy), we suggest using early stopping. Starting from the optimal model trained on the dataset $\mathcal{D}_i$ of party $i$ (with the performance $v(i)/\nu = s(i)$), we continue to train the model with the aggregated dataset $\mathcal{D}_\mathcal{N}$ and stop the training when the performance of the model reaches $(v(i) + \lambda_i^* p_{i,-}(v))/\nu$.

*Remark* 4.2. There always exists a party in $\mathcal{N}$ that receives the optimal model trained on $\mathcal{D}_\mathcal{N}$ (of value $v(\mathcal{N})$). Unlike the work of [16] enforcing this property, namely *weak efficiency*, from the construction of the allocation scheme, this property naturally arises from our above fairness properties as shown in App. H. It implies that all of the aggregated dataset is utilized by at least a party.

*Remark* 4.3. Our proposed framework can be extended to handle the performance constraint. In particular, while the budget constraint prevents a party's expense from exceeding its budget, the performance constraint implies that a party is not interested in obtainng a model with a performance larger than a *desirable model performance*. The details are in App. I.

# 5 Experiments

This section empirically illustrates the proposed allocation scheme of both payoff and model rewards in collaborative ML. To easily interpret the results, we choose the MNIST dataset [10] which contains $70,000$ images of handwritten digits. The dataset is partitioned based on the digit labels. A party, denoted as $[s\text{-}e]$ ($s \leq e$), owns a subset of the MNIST training dataset (of size $60,000$) that are labeled with digits $s, s + 1, \ldots, e$. There are 5 parties in $\mathcal{N}$: $[0]$ (i.e., [0-0]), [1-2], [0-3], [3-5], and [6-9]. For simplicity, we set $\nu = 1$. The model performance is measured by the prediction accuracy on a separate dataset of size $10,000$.

Additional experiments on the CIFAR-10 [9] dataset and the IMDB movie reviews dataset [11] dataset are in App. J. All experiments share the following 4 desirable observations. **First**, when there are not any budget constraints, all parties receive models with the highest performance, and parties with more valuable datasets receive higher payoff rewards. **Second**, even if all parties' budgets are 0, they still benefit from the collaboration (hence, motivated to join the collaboration) by obtaining better models than their own. **Third**, increasing the budget of a party not only increases the model performance of the party but can also benefit other parties through improved model performance and/or higher payoffs. **Fourth**, there is always a party receiving the best model of value $v(\mathcal{N})$, i.e., the weak efficiency property.

Figs. 2a and 2b show the payoff and model rewards when there are not any budget constraints (i.e., $\mathbf{b} = \infty \mathbf{1}$), respectively. In Fig. 2a, party [6-9] receives the largest payoff reward because party [6-9] owns the dataset with the largest number of digits which is not overlapped with any other parties' datasets. Party [0-3] receives a larger payoff reward than that of parties [0] and [1-2] because party [0-3]'s dataset contains the datasets of both parties [0] and [1-2]. In Fig. 2b, all parties receive models with the highest performance as there are not any budget constraints. While the work of [13] can address the case of no budget constraints, it results in a payoff vector $[-0.41, -0.26, -0.02, 0.12, 0.58]$ which is different from ours. It is noted that their solution does not satisfy the linear property while ours does.

Figs. 2c and 2d show the payoff and model rewards when all parties are not allowed to spend any payoff (i.e, $\mathbf{b} = \mathbf{0}$), respectively. From the balance property, the payoff rewards are 0 for all parties (in Fig. 2c). However, in Fig. 2d, they still benefit from the collaboration by obtaining better models (of value $v(i) + p_{i,-}(\lambda_i v)$) than their own models (of value $v(i)$). In particular, we observe that party [6-9] receives the optimal model trained on the aggregated dataset $\mathcal{D}_\mathcal{N}$ (of value $v(\mathcal{N})$), as stated by the weak efficiency property. Thus, when monetary payoffs are not allowed, our allocation scheme is still able to incentivize parties to fairly collaborate using their data. Compared to the $\rho$-Shapley

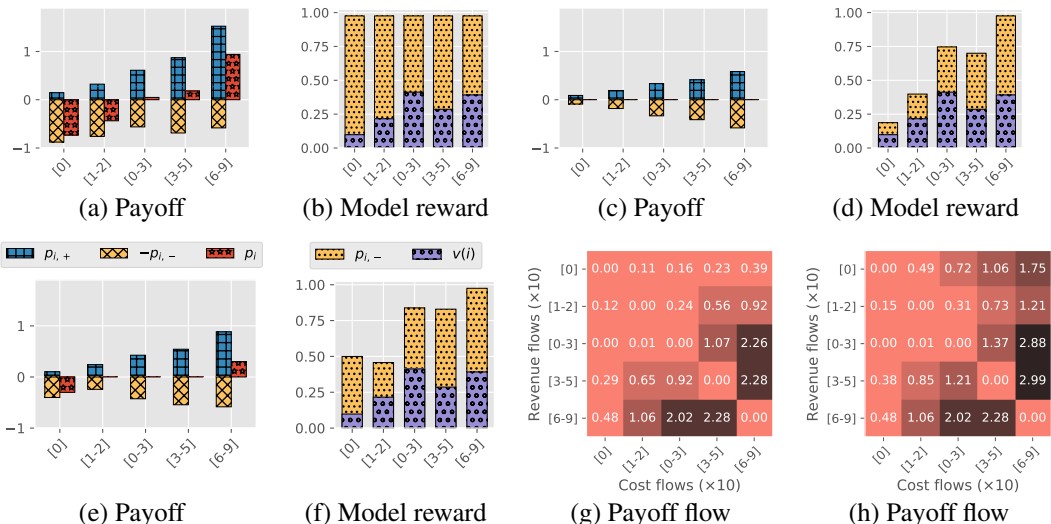

Figure 2: Plots of the payoffs, model rewards, and payoff flows in the MNIST experiments with budget constraints: (a,b) $\mathbf{b} = \infty\mathbf{1}$, (c,d,g) $\mathbf{b} = \mathbf{0}$, and (e,f,h) $b_i = 0.3\mathbb{1}_{i=1}$.

value [16] (which only works in this special case), the model reward allocation in Fig. 2d cannot be obtained from the $\rho$-Shapley value (by varying its parameters). In fact, unlike our proposed allocation scheme, when $\rho \neq 1$, the $\rho$-Shapley value does not satisfy the linear property.

Figs. 2e and 2f show the payoff and model rewards when only party [0] has a positive budget of $0.3$ (other parties have budgets of 0), i.e., $b_i = 0.3\mathbb{1}_{i=1}$. While it is intuitive that with a positive budget, party [0] improves the performance of its obtained model in comparison to the previous case of no budget, it is interesting that parties [1-2], [0-3], and [3-5] also have the performance of their models improved (by comparing Figs. 2f and 2d). The reason is that revenues of parties [1-2], [0-3], and [3-5] from party [0] increase (comparing the first row in Figs. 2g and 2h). These increased revenues are used to further improve the models of [1-2], [0-3], and [3-5], e.g., by getting more contributions from party [6-9] (comparing the last column in Figs. 2g and 2h). Furthermore, not only does party [6-9] obtain the best model of value $v(\mathcal{N})$ as that in Fig. 2d, but it also obtains an additional payoff reward for its contributions to other parties in Fig. 2e. As seen in Figs. 2g and 2h, the cost flows of party [6-9] (the last rows) are the same between the two cases. Thus, according to our allocation scheme, increasing the budget of a party benefits not only itself but also other parties.

# 6   Conclusion

This paper presents an allocation scheme for both payoff and model rewards in fair collaborative ML. When there are not any budget constraints, all parties receive the same model trained on the aggregated dataset. On the other hand, they receive different payoff rewards to account for the difference in the contributions of different parties. We construct such a payoff allocation scheme that is fair according to (i) desirable properties of the parties' payoffs or (ii) that of the underlying payoff flows from one party to another. The latter is useful to handle the practical constraint on the budgets of parties. In particular, when there exist budget constraints, we propose a solution to trade off between payoff and model rewards through fair adjustments of the payoff flows. The experiments empirically show that our scheme is reasonable in several collaborative ML scenarios with different budget constraints.

## Acknowledgments and Disclosure of Funding

This research/project is supported by the National Research Foundation Singapore and DSO National Laboratories under the AI Singapore Programme (AISG Award No: AISG2-RP-2020-018).

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
