## A Brief Review of The Shapley Value

Given a value function $v$, the Shapley value is a solution to distributing the payoff $v(\mathcal{N})$ to parties in $\mathcal{N}$ [14]. Let us imagine that the grand coalition is formed by one party joining the coalition at a time. Given an order of parties (i.e., a permutation $\pi$ of $\mathcal{N}$), party $i$ joins the coalition $\mathcal{P}_\pi^i$ which denotes all parties preceding $i$ in $\pi$. Then, the *marginal contribution* of party $i$ to $\mathcal{P}_\pi^i$ is $\Delta_v(i|\mathcal{P}_\pi^i) \triangleq v(\mathcal{P}_\pi^i \cup i) - v(\mathcal{P}_\pi^i)$. The Shapley value of party $i$, denoted as $\varphi_i(v)$, is the average of the marginal contributions over the set $\Pi$ of all possible permutations:

$$\varphi_i(v) \triangleq \frac{1}{n!} \sum_{\pi \in \Pi} \Delta_v(i|\mathcal{P}_\pi^i) \,. \tag{12}$$

Alternatively, it is also evaluated as

$$\varphi_i(v) = \frac{1}{n} \sum_{\mathcal{C} \in \mathcal{N} \setminus i} \binom{n-1}{|\mathcal{C}|}^{-1} \Delta_v(i|\mathcal{C})$$

where $\binom{n-1}{|\mathcal{C}|}$ denotes the number of combinations (the selections) of $|\mathcal{C}|$ items from a set of $n-1$ distinct items. The Shapley value is 'fair' since it is the unique solution that satisfies several desirable properties as elaborated below.

**Efficiency.** $\sum_i \varphi_i(v) = v(\mathcal{N})$. It ensures that all of $v(\mathcal{N})$ are distributed to the parties.

**Symmetry.** If $v(\mathcal{C} \cup i) = v(\mathcal{C} \cup j)$ for all $\mathcal{C} \subset \mathcal{N} \setminus \{i, j\}$, then $\varphi_i(v) = \varphi_j(v)$. It implies parties with equal marginal contributions to any coalitions have the same payoff.

**Dummy party.** A *dummy party* $i \in \mathcal{N}$ satisfies $v(\mathcal{C} \cup i) = v(\mathcal{C})$ for all $\mathcal{C} \subset \mathcal{N} \setminus i$. For a dummy party $i$, $\varphi_i(v) = 0$.

**Linearity.** Let us consider the value function as an element in a real vector space of dimension $2^n$ (i.e., the number of subsets of $\mathcal{N}$) which is equipped with the usual addition and scalar multiplication. Then, given 2 value functions $v, w$, and a real number $\alpha \in \mathbb{R}$,

$$\varphi_i(v + w) = \varphi_i(v) + \varphi_i(w) \,, \quad \varphi_i(\alpha v) = \alpha \varphi_i(v) \,.$$

## B Proof of Lemma 3.1

Let us consider the value function $v'$ such that $v'(\mathcal{C}) = nv(\mathcal{C})$ for all $\mathcal{C} \subset \mathcal{N}$. Given $n > 0$, it is noted that $v'$ is uniquely defined given $v$ and vice versa.

Let us define the function $\zeta_i(v') \triangleq p_i(v) + v(\mathcal{N})$.

First, from the linearity and symmetry properties of $p_i(v)$ and the definition of $v'$, it follows that $\zeta_i(v')$ satisfies the linearity and the symmetry properties.

Second, we observe that

$$\sum_{i \in \mathcal{N}} \zeta_i(v') = \sum_{i \in \mathcal{N}} (p_i(v) + v(\mathcal{N})) = nv(\mathcal{N}) = v'(\mathcal{N}) \,.$$

Hence, $\zeta_i(v')$ satisfies the efficiency property w.r.t. the value function $v'$.

Third, if $i$ is a dummy party w.r.t. $v$, then $i$ is also a dummy party w.r.t. $v'$. Furthermore,

$$\zeta_i(v') = p_i(v) + v(\mathcal{N}) = -v(\mathcal{N}) + v(\mathcal{N}) = 0 \,.$$

Hence, $\zeta_i(v')$ satisfies the dummy property w.r.t. $v'$.

Therefore, from the uniqueness of the Shapley value, $\zeta_i(v')$ is uniquely defined as the Shapley value w.r.t. $v'$, i.e.,

$$\zeta_i(v') = \varphi_i(v') = n\varphi_i(v)$$

where we make use of the linearity property of the Shapley value in the second equality.

As a result, $p_i(v)$ is uniquely defined as

$$p_i(v) = n\varphi_i(v) - v(\mathcal{N}) \,.$$

# C  Proof of Remark 3.6

Let us consider the value function $v'$ such that $v'(\mathcal{C}) = \alpha v(\mathcal{C})$ for all $\mathcal{C} \subset \mathcal{N}$. Given $n > 0$, it is noted that $v'$ is uniquely defined given $v$ and vice versa.

Let us define the function

$$\zeta_i(v') \triangleq p_i(v) + \frac{\alpha}{n} v(\mathcal{N}) - \frac{\beta}{n} \, . \tag{13}$$

First, from the linearity and symmetry properties of $p_i(v)$ and the definition of $v'$, it follows that $\zeta_i(v')$ satisfies the linearity and the symmetry properties.

Second, we observe that

$$\sum_{i \in \mathcal{N}} \zeta_i(v') = \sum_{i \in \mathcal{N}} \left( p_i(v) + \frac{\alpha}{n} v(\mathcal{N}) - \frac{\beta}{n} \right) = \alpha v(\mathcal{N}) = v'(\mathcal{N}) \, .$$

Hence, $\zeta_i(v')$ satisfies the efficiency property w.r.t. the value function $v'$.

Third, if $i$ is a dummy party w.r.t. $v$, then $i$ is also a dummy party w.r.t. $v'$. Furthermore,

$$\zeta_i(v') = p_i(v) + \frac{\alpha}{n} v(\mathcal{N}) - \frac{\beta}{n} = 0 \, .$$

Hence, $\zeta_i(v')$ satisfies the dummy property w.r.t. $v'$.

Therefore, from the uniqueness of the Shapley value, $\zeta_i(v')$ is uniquely defined as the Shapley value w.r.t. $v'$, i.e.,

$$\zeta_i(v') = \varphi_i(v') = \alpha \varphi_i(v)$$

where we make use of the linearity property of the Shapley value in the second equality.

As a result, $p_i(v)$ is uniquely defined as

$$p_i(v) = \alpha \varphi_i(v) - \frac{\alpha}{n} v(\mathcal{N}) + \frac{\beta}{n} \, .$$

# D  Replication Robustness

A reward allocation scheme is *replication-robust* if a party cannot increase its rewards by replicating its data and participating in the collaboration as multiple parties.

It is well-known that the Shapley value, despite its fairness, is not replication robustness in data valuation [1]. This is because the two desirable properties for fairness: symmetry and efficiency violate the replication robustness. Let us consider the example in the work of [1]: the grand coalition of 2 parties $\mathcal{N} = \{1, 2\}$ where party 1 and 2 have the same marginal contributions to any coalition $\mathcal{C} \subset \mathcal{N}$. Suppose the value of the grand coalition $\mathcal{N}$ is $v(\mathcal{N}) = 1$. Due to the efficiency and symmetry properties of the Shapley value, $\varphi_1(v) = \varphi_2(v) = 1/2$. Then, suppose party 1 replicates itself into another party $1^+$ (i.e., $\mathcal{D}_1 = \mathcal{D}_{1+}$), the new grand coalition is $\mathcal{N}^+ = \{1, 1^+, 2\}$. Let $v^+$ denote the new value function for coalitions in $\mathcal{N}^+$. Then, $v^+(\mathcal{N}^+) = v(\mathcal{N}) = 1$. Due to the efficiency and symmetry properties of the Shapley value, $\varphi_1(v^+) = \varphi_{1+}(v^+) = \varphi_2(v^+) = 1/3$. Hence, the total contributions of party 1 and its replication $1^+$ is $\varphi_1(v^+) + \varphi_{1+}(v^+) = 2/3 > \varphi_1(v)$. Therefore, a data valuation according to the Shapley value is not replication-robust.

While the Shapley value can be customized to be replication-robust [6], the efficiency property does not hold for the customized variant.

In this work, we are interested in maintaining both the efficiency and the symmetry properties of an allocation scheme. Thus, replication-robustness is sacrificed. However, as we elaborate in the following paragraphs, this property still holds for some parties (e.g., parties with negative payoff rewards) under the payoff allocation scheme in Lemma 3.1.

## D.1 One Replication

A party is a *replication* of another party if their training datasets are the same. Let us consider the case that in the grand coalition $\mathcal{N}^+$, there exists a party $i^+ \in \mathcal{N}$ that is a replication of another party $i \in \mathcal{N} \setminus i^+$ (i.e., $\mathcal{D}_i = \mathcal{D}_{i+}$). Let $\mathcal{N} = \mathcal{N}^+ \setminus i^+$ be the coalition that does not contain the replication $i^+$.

Let $v$ and $v^+$ denote the value functions of coalitions in $\mathcal{N}$ and $\mathcal{N}^+$, respectively. We assume that the replication does not change the value of the coalition, i.e., for all $\mathcal{C} \subset \mathcal{N} \setminus i$,

$$
\begin{aligned}
v^+(\mathcal{C} \cup \{i, i^+\}) = v^+(\mathcal{C} \cup i) = v^+(\mathcal{C} \cup i^+) \quad &= v(\mathcal{C} \cup i) \\
v^+(\mathcal{C}) \quad &= v(\mathcal{C})
\end{aligned}
\tag{14}
$$

We are interested in the condition where party $i$ is discouraged from replicating itself: the total payoff reward of both $i$ and $i^+$ in $\mathcal{N}^+$ is smaller than the payoff reward of $i$ in $\mathcal{N}$, i.e.,

$$
\begin{aligned}
p_i(v^+) + p_{i+}(v^+) &\leq p_i(v) \\
2p_i(v^+) &\leq p_i(v) \tag{15} \\
2(n+1)\varphi_i(v^+) - 2v(\mathcal{N}^+) &\leq n\varphi_i(v) - v(\mathcal{N}) \\
\varphi_i(v^+) &\leq \frac{n\varphi_i(v) + v(\mathcal{N})}{2(n+1)} \tag{16}
\end{aligned}
$$

where (15) is due to the symmetry property of the payoff reward; and (16) is due to $v(\mathcal{N}^+) = v(\mathcal{N})$ (14).

The Shapley value $\varphi_i(v^+)$ of party $i$ in $\mathcal{N}^+$ is computed as follows.

$$
\begin{aligned}
&\varphi_i(v^+) \\
&= \frac{1}{n+1} \sum_{\mathcal{C} \subset \mathcal{N}^+ \setminus i} \binom{n}{|\mathcal{C}|}^{-1} \Delta_v(i|\mathcal{C}) \\
&= \frac{1}{n+1} \sum_{\mathcal{C} \subset \mathcal{N}^+ \setminus i} \binom{n}{|\mathcal{C}|}^{-1} \mathbb{1}_{i^+ \notin \mathcal{C}} \Delta_v(i|\mathcal{C}) \tag{17} \\
&= \frac{1}{n+1} \sum_{\mathcal{C} \subset \mathcal{N}^+ \setminus \{i, i^+\}} \binom{n}{|\mathcal{C}|}^{-1} \Delta_v(i|\mathcal{C}) \\
&= \frac{1}{n} \sum_{\mathcal{C} \subset \mathcal{N} \setminus i} \frac{n - |\mathcal{C}|}{n+1} \binom{n-1}{|\mathcal{C}|}^{-1} \Delta_v(i|\mathcal{C}) \tag{18}
\end{aligned}
$$

where (17) is due to (14).

From (16), (18), and the definition of the Shapley value, the condition to discourage party $i$ from replicating itself is

$$
n\varphi_i(v) - \sum_{\mathcal{C} \in \mathcal{N} \setminus i} \frac{2|\mathcal{C}|}{n} \binom{n-1}{|\mathcal{C}|}^{-1} \Delta_v(i|\mathcal{C}) \leq v(\mathcal{N}) .
\tag{19}
$$

Since $\Delta_v(i|\mathcal{C}) \geq 0$ (the monotonicity property), $n\varphi_i(v) \leq v(\mathcal{N})$ implies the above condition. From the Corollary 3.2, it means that a party with a negative payoff reward cannot improve its payoff reward by replicating itself.

## D.2 Multiple Replications

In this section, we would like to show that: if a party cannot improve its payoff reward by replicating itself once, then it cannot improve its payoff reward regardless of the number of its replications. Hence, the condition (19) also applies to the case of multiple replications.

First, we can show that the payoff reward of party $i$ reduces if it replicates itself once, i.e., $p_i(v^+) \leq p_i(v)$. From (18),

$$\varphi_i(v^+) = \frac{1}{n} \sum_{\mathcal{C} \subset \mathcal{N} \setminus i} \frac{n - |\mathcal{C}|}{n + 1} \binom{n-1}{|\mathcal{C}|}^{-1} \Delta_v(i|\mathcal{C})$$

$$\leq \frac{1}{n} \sum_{\mathcal{C} \subset \mathcal{N} \setminus i} \frac{n}{n + 1} \binom{n-1}{|\mathcal{C}|}^{-1} \Delta_v(i|\mathcal{C})$$

$$= \frac{n}{n + 1} \varphi_i(v) .$$

Hence,

$$(n + 1)\varphi_i(v^+) \leq n\varphi_i(v)$$
$$(n + 1)\varphi_i(v^+) - v(\mathcal{N}^+) \leq n\varphi_i(v) - v(\mathcal{N})$$
$$p_i(v^+) \leq p_i(v) . \tag{20}$$

Let us consider party $i$ that cannot improve its payoff reward by replicating once. We can view the case that there are $K$ replications of party $i$ as party $i$ replicates itself $K$ times sequentially. Suppose after $k$ replications, the grand coalition $\mathcal{N}$ contains $k$ replications of party $i$ (excluding party $i$). Let $\mathcal{N}^+$ be the grand coalition after party $i$ replicates itself once more time, i.e., $\mathcal{N}^+ = \mathcal{N} \cup i^+$ where $i^+$ is a new replication of party $i$. In other words, there are $k + 1$ replications of party $i$ in $\mathcal{N}^+$ (excluding party $i$). It is noted that from $\mathcal{N}$ to $\mathcal{N}^+$, party $i$ replicates itself once, so the results in Sec. D.1 apply. Let $v$ and $v^+$ be the same notations in Sec. D.1. The total payoff rewards of all $k + 1$ replications of party $i$ and party $i$ in $\mathcal{N}^+$ is:

$$\underbrace{(k + 1)p_i(v^+)}_{\text{payoff of } k \text{ replications in } \mathcal{N} \text{ and } i} + \underbrace{p_{i^+}(v^+)}_{\text{payoff of the new replication } i^+}$$

$$\leq (k + 1)p_i(v) + p_{i^+}(v^+) \quad \text{from (20)}$$
$$= kp_i(v) + (p_i(v) + p_{i^+}(v^+))$$
$$\leq kp_i(v) + p_i(v) \tag{21}$$
$$= (k + 1)p_i(v)$$

where (21) is because we assume that party $i$ cannot improve its payoff reward by replicating itself once. Hence, from $\mathcal{N}$ to $\mathcal{N}^+$ the total payoff rewards of all replications of party $i$ and party $i$ do not improve. Applying this argument for $K$ replications of party $i$ sequentially, we obtain the result that: if a party cannot improve its payoff reward by replicating itself once, then it cannot improve its payoff reward regardless of the number of its replications.

# E   Proof of Lemma 3.7

Let us define a value function $v_{-i}$ on $\mathcal{N}_{-i} \triangleq \mathcal{N} \setminus i$ such that for all $\mathcal{C} \subset \mathcal{N}_{-i}$,

$$v_{-i}(\mathcal{C}) = v(\mathcal{C} \cup i) - v(i) .$$

Then,

$$v_{-i}(\emptyset) = 0$$
$$\Delta_{v_{-i}}(j|\mathcal{C}) = \Delta_v(j|\mathcal{C} \cup i) . \tag{22}$$

We observe that the 4 properties we propose for payoff flows $(p_{ij})_{j \in \mathcal{N} \setminus i}$ in Sec. 3.2 (i.e., fair cost, conditional symmetry, linearity, and conditional dummy party) corresponds to the 4 properties: efficiency, symmetry, linearity, and dummy party defined with $v_{-i}$ in app. A as follows.

Firstly, the fair cost implies that

$$\sum_{j \in \mathcal{N} \setminus i} p_{ij}(v) = v(\mathcal{N}) - v(i) = v_{-i}(\mathcal{N} \setminus i)$$

which is the efficiency property defined on $\mathcal{N}_{-i}$ with $v_{-i}$.

Secondly, from (22), the conditional symmetry property is equivalent to the following symmetry property defined with $v_{-i}$: if two parties $j, k \in \mathcal{N}_{-i}$ have the same marginal contribution to any coalition $\mathcal{C} \cup i$ for $\mathcal{C} \subset \mathcal{N}_{-i} \setminus \{j, k\}$, i.e.,

$$\Delta_{v_{-i}}(j|\mathcal{C}) = \Delta_{v_{-i}}(k|\mathcal{C}) \,,$$

then

$$p_{ij} = p_{ik} \,.$$

Thirdly, the linearity property in Sec. 3.2 is equivalent to that in App. A.

Fourthly, the conditional dummy party property is equivalent to the following dummy party property defined with $v_{-i}$: a conditional dummy party $j \in \mathcal{N}_{-i}$ given party $i$ satisfies $v_{-i}(\mathcal{C} \cup j) = v_{-i}(\mathcal{C})$ for all $\mathcal{C} \subset \mathcal{N}_{-i} \setminus j$ is a dummy party w.r.t. $v_{-i}$ on $\mathcal{N}_{-i}$.

Therefore, the unique solution that satisfies the above 4 properties is the Shapley value defined on $\mathcal{N}_{-i}$ with $v_{-i}$: for all $j \in \mathcal{N}_{-i}$,

$$\varphi_j(v_{-i}) = \frac{1}{n-1} \sum_{\mathcal{C} \in \mathcal{N}_{-i} \setminus j} \binom{n-2}{|\mathcal{C}|}^{-1} \Delta_{v_{-i}}(j|\mathcal{C})$$

$$= \frac{1}{n-1} \sum_{\mathcal{C} \in \mathcal{N}_{-i} \setminus j} \binom{n-2}{|\mathcal{C}|}^{-1} \Delta_v(j|\mathcal{C} \cup i) \tag{23}$$

where (23) is due to (22). This defines the conditional Shapley value $\varphi_{j|i}(v)$ given party $i$, i.e., $\varphi_{j|i}(v) = \varphi_j(v_{-i})$. Alternatively, it can be evaluated as:

$$\varphi_{j|i}(v) = \frac{1}{(n-1)!} \sum_{\pi \in \Pi_i} \Delta_v(j|\mathcal{P}_\pi^j) \tag{24}$$

where $\Pi_i$ denotes the set of permutations that start with $i$ (i.e., $i$ is at the first position of $\pi \in \Pi_i$).

## F   Decomposition of The Shapley Value

Recall in (12), the Shapley value can be evaluated as follows

$$\varphi_i(v) \triangleq \frac{1}{n!} \sum_{\pi \in \Pi} \Delta_v(i|\mathcal{P}_\pi^i)$$

where $\Pi$ is the set of all permutations of $\mathcal{N}$.

Let $\Pi$ be partitioned based on the first party of $\pi \in \Pi$: $\Pi = \cup_{i \in \mathcal{N}} \Pi_i$ where $\Pi_i$ denote the set of permutations that start with $i$.

$$\varphi_i(v) = \frac{1}{n!} \sum_{j \in \mathcal{N}} \sum_{\pi \in \Pi_j} \Delta_v(i|\mathcal{P}_\pi^i)$$

$$= \frac{1}{n} \sum_{j \in \mathcal{N}} \frac{1}{(n-1)!} \sum_{\pi \in \Pi_j} \Delta_v(i|\mathcal{P}_\pi^i)$$

Furthermore,

$$|\Pi_j| = (n-1)! \,, \forall j \in \mathcal{N}$$
$$\Delta_v(i|\mathcal{P}_{\pi_i}^i) = \Delta_v(i|\emptyset) = v(i)$$

Therefore,

$$\varphi_i(v) = \frac{1}{n} \sum_{j \in \mathcal{N}} \frac{1}{(n-1)!} \sum_{\pi \in \Pi_j} \Delta_v(i|\mathcal{P}_\pi^i)$$

$$= \frac{1}{n} \left( v(i) + \sum_{j \in \mathcal{N} \setminus i} \frac{1}{(n-1)!} \sum_{\pi \in \Pi_j} \Delta_v(i|\mathcal{P}_\pi^i) \right)$$

$$= \frac{1}{n} \left( v(i) + \sum_{j \in \mathcal{N} \setminus i} \varphi_{i|j}(v) \right)$$

where the last equality is due to (24). Hence, we obtain (6).

## G   Proof of Lemma 4.1

In this section, we prove that the feasibility & individual rationality, the marginal contribution consistency, and the budget efficiency uniquely determine an adjusted payoff flows as the conditional Shapley value of an adjusted value function. Hence, the 4 fairness properties in Sec. 3.2 still hold.

The outline of the proof of Lemma 4.1 is as follows.

We first show that the feasibility & individual rationality and the marginal contribution consistency properties uniquely determine adjusted payoff flows (up to the choice of scaling parameters) in Lemma G.1 in App. G.1. Given these adjusted payoff flows, proving Lemma 4.1 boils down to proving that the scaling parameters are uniquely defined by the budget constraints and the budget efficiency as the optimal solution to (P) which requires proving the existence and the uniqueness of the scaling parameters.

For the existence proof, we show that (P) always has a finite optimal solution, and when the scaling parameters in Lemma G.1 are chosen as this optimal solution, the adjusted payoff flows satisfy the budget constraints and the budget efficiency in App. G.2.

For the uniqueness proof, we show that the budget constraints and the budget efficiency uniquely define the scaling parameters in App. G.3.

Additionally, we prove that the payoff flows also satisfy the weak efficiency property in App. H.

### G.1   Adjusted Payoff Flows from Feasibility & Individual Rationality and Marginal Contribution Consistency

**Lemma G.1.** *The feasibility & individual rationality and the marginal contribution consistency properties uniquely determine the following adjusted value function (up to the choice of scaling parameters $\lambda_i \in [0,1]$):*

$$v_i'(\mathcal{C} \cup i) = \lambda_i v(\mathcal{C} \cup i) + (1 - \lambda_i)v(i) \tag{25}$$

*where $\lambda_i \in [0,1]$ for all $\mathcal{C} \in \mathcal{N} \setminus i$ and $i \in \mathcal{N}$.*

*Proof.* Let $v'$ denote the value function that satisfies both the feasibility & individual rationality and the marginal contribution consistency properties. From the feasibility & individual rationality,

$$v(\mathcal{C} \cup i) \geq v_i'(\mathcal{C} \cup i) \geq v(i) .$$

Choose $\mathcal{C} = \emptyset$, then

$$v_i'(i) = v(i) . \tag{26}$$

From the marginal contribution consistency, $\exists \lambda_i > 0, \forall j \in \mathcal{N} \setminus i, \forall \mathcal{C} \subset \mathcal{N} \setminus \{i, j\}$,

$$\Delta_{v_i'}(j|\mathcal{C} \cup i) = \lambda_i \Delta_v(j|\mathcal{C} \cup i)$$

$$v_i'(\mathcal{C} \cup \{i, j\}) - v_i'(\mathcal{C} \cup i) = \lambda_i \left( v(\mathcal{C} \cup \{i, j\}) - v(\mathcal{C} \cup i) \right) \tag{27}$$

Consider a coalition $\mathcal{C} \subset \mathcal{N} \setminus i$, let $\pi$ be a permutation of parties in $\mathcal{C}$ and $\pi(j)$ denote the party at the $j$-th position in $\pi$. Then,

$$v_i'(\mathcal{C} \cup i) - v_i'(i)$$

$$= \sum_{j=1}^{|\mathcal{C}|} v_i'(\mathcal{P}_\pi^{\pi(j)} \cup \{\pi(j), i\}) - v_i'(\mathcal{P}_\pi^{\pi(j)} \cup i)$$

$$= \sum_{j=1}^{|\mathcal{C}|} \lambda_i \left( v(\mathcal{P}_\pi^{\pi(j)} \cup \{\pi(j), i\}) - v(\mathcal{P}_\pi^{\pi(j)} \cup i) \right)$$

$$= \lambda_i \sum_{j=1}^{|\mathcal{C}|} v(\mathcal{P}_\pi^{\pi(j)} \cup \{\pi(j), i\}) - v(\mathcal{P}_\pi^{\pi(j)} \cup i)$$

$$= \lambda_i \left( v(\mathcal{C} \cup i) - v(i) \right)$$

where $\mathcal{P}_\pi^{\pi(j)}$ denote the set of all parties preceding $\pi(j)$ in the permutation $\pi$ (i.e., the set of parties with the positions less than $j$ in $\pi$). Furthermore, since $v(i) = v_i'(i)$ (26),

$$v_i'(\mathcal{C} \cup i) - v(i) = \lambda_i \left( v(\mathcal{C} \cup i) - v(i) \right)$$
$$v_i'(\mathcal{C} \cup i) = \lambda_i v(\mathcal{C} \cup i) + (1 - \lambda_i) v(i) . \tag{28}$$

For (28) to satisfy the feasibility & individual rationality, $\lambda_i \in [0, 1]$.

On the other hand, one can also verify that (28) for $\lambda_i \in [0, 1]$ satisfies both the feasibility & individual rationality and the marginal contribution consistency properties. Therefore, it is the unique solution (up to the choice of $\lambda_i$). $\square$

**Lemma G.2.** *Given the adjusted value functions $v_i'$, the adjusted payoff flows from party $i$ is uniquely defined through the conditional Shapley value:*

$$\forall j \in \mathcal{N} \setminus i, \; p_{ij}(v_i') = \varphi_{j|i}(v_i') = \varphi_{j|i}(\lambda_i v) = \lambda_i \varphi_{j|i}(v) . \tag{29}$$

*Proof.* Let us define $\tilde{v}$ as follows

$$\tilde{v}(\mathcal{C} \cup i) = v'(\mathcal{C} \cup i) - (1 - \lambda_i) v'(i), \; \forall \mathcal{C} \subset \mathcal{N} \setminus i \tag{30}$$

Due to the translation invariance of the conditional Shapley value in Corollary 3.8,

$$\varphi_{j|i}(v') = \varphi_{j|i}(\tilde{v}) . \tag{31}$$

Furthermore,

$$\tilde{v}(\mathcal{C} \cup i) = v'(\mathcal{C} \cup i) - (1 - \lambda_i) v'(i) \tag{32}$$
$$= \lambda_i v(\mathcal{C} \cup i) \tag{33}$$

Hence, due to the linearity property of the Shapley value,

$$\varphi_{j|i}(\tilde{v}) = \varphi_{j|i}(\lambda_i v) = \lambda_i \varphi_{j|i}(v) . \tag{34}$$

$\square$

Next, given that $p_{ij}(v_i') = \lambda_i \varphi_{j|i}(v)$ for all $\{i, j\} \subset \mathcal{N}$, we will prove that the value of $\lambda_i \in [0, 1]$ is uniquely determined from the budget constraints and the budget efficiency as the optimal solution $\boldsymbol{\lambda}^*$ to (P).

## G.2 The Existence of $\lambda$

We observe that (P) always has a feasible solution of $\boldsymbol{\lambda} = \mathbf{0}$ and its objective function is bounded:

$$\sum_{\{i,j\} \subset \mathcal{N}} \lambda_i \varphi_{j|i}(v) \leq \sum_{\{i,j\} \subset \mathcal{N}} \varphi_{j|i}(v)$$

since $\lambda_i \in [0, 1]$ for all $i \in \mathcal{N}$. Therefore, (P) always has a finite optimal solution, denoted as $\boldsymbol{\lambda}^* = (\lambda_i^*)_{i=1}^n$. Then, we will prove that the payoff flows specified by $p_{ij}(v_i') = \lambda_i^* \varphi_{j|i}(v)$ satisfy the budget constraints and the budget efficiency property as follows.

**Budget constraints.** As discussed in Sec. 4, the total revenue flow and the total cost flow by distributing the payoff according to the payoff flows $p_{ij}(v_i') = \lambda_i^* \varphi_{j|i}(v)$ are

$$p_{i,+}(\boldsymbol{\lambda}^* \vec{v}) = \sum_{j \in \mathcal{N} \setminus i} \lambda_j^* \varphi_{i|j}(v) \text{ and } p_{i,-}(\boldsymbol{\lambda}^* \vec{v}) = p_{i,-}(\lambda_i^* v) = \lambda_i^* \sum_{j \in \mathcal{N} \setminus i} \varphi_{j|i}(v) ,$$

respectively. Thus, the LHS of the first set of constraints in (P) are $-p_i(\boldsymbol{\lambda} \vec{v}) = p_{i,-}(\boldsymbol{\lambda} \vec{v}) - p_{i,+}(\boldsymbol{\lambda} \vec{v})$, $\forall i \in \mathcal{N}$, so they are the budget constraints. Therefore, any feasible solution (including the optimal $\boldsymbol{\lambda}^*$) to (P) produces payoff flows that satisfy the budget constraints.

**Budget efficiency.** We show that the payoff flows $p_{ij}(v_i') = \lambda_i^* \varphi_{j|i}(v)$ for all $\{i, j\} \subset \mathcal{N}$ satisfy the budget efficiency by proving the following two statements.

- **Statement 1:** If party $i$ can afford the optimal model of value $v(\mathcal{N})$ (i.e., $-p_i(v) \leq b_i$), it receives the optimal model of value $v(\mathcal{N})$ under the adjusted payoff flows, i.e., $\lambda_i^* = 1$.

- **Statement 2:** If party $i$ cannot afford the optimal model of value $v(\mathcal{N})$ due to its budget $-p_i(v) > b_i$, then $-p_i(\vec{v}') = b_i$.

**Statement 1:** To prove by contradiction, assuming that party $i$ can afford the optimal model of value $v(\mathcal{N})$ (i.e., $-p_i(v) \leq b_i$) and $\lambda_i^* < 1$. Let us define $\boldsymbol{\lambda}' \triangleq (\lambda_j')_{j=1}^n$ as follows

$$\begin{cases} \lambda_i' = 1 \\ \lambda_j' = \lambda_j^* \text{ for } j \in \mathcal{N} \setminus i \end{cases}$$

It is noted that

$$\begin{aligned} p_{i,-}(\lambda_i' v) - p_{i,+}(\boldsymbol{\lambda}' v) = p_{i,-}(v) - \sum_{j \in \mathcal{N} \setminus i} \lambda_j^* \varphi_{i|j}(v) &\geq p_{i,-}(v) - \sum_{j \in \mathcal{N} \setminus i} \varphi_{i|j}(v) \\ &\geq p_{i,-}(v) - p_{i,+}(v) \\ &= p_i(v) \\ &\geq -b_i . \end{aligned}$$

Hence, $\boldsymbol{\lambda}'$ satisfies the budget constraint of party $i$. Furthermore, it also satisfies the budget constraints of other parties in $\mathcal{N} \setminus i$ since increasing $\lambda_i$ from $\lambda_i^* < 1$ to 1 only increases the revenue of other parties in $\mathcal{N} \setminus i$. Therefore, $\boldsymbol{\lambda}'$ is a feasible solution to (P). It also has a higher objective function (since $\lambda_i$ increases) than the optimal solution $\boldsymbol{\lambda}^*$, which is a contradiction. Therefore, $\lambda_i^* = 1$ which implies that the value of the model that party $i$ receives under the adjusted value function $\vec{v}' \triangleq (\lambda_i^* v)_{i=1}^n$ is $v(i) + p_{i,-}(v) = v(\mathcal{N})$.

**Statement 2:** To prove by contradiction, assuming that there exists a party $i$ such that

$$-p_i(\boldsymbol{\lambda}^* v) < b_i < -p_i(v) . \tag{35}$$

Furthermore,

$$\begin{aligned} -p_i(\boldsymbol{\lambda}^* v) = \lambda_i^* p_{i,-}(v) - \sum_{j \in \mathcal{N} \setminus i} \lambda_j^* \varphi_{i|j}(v) \\ \geq \lambda_i^* p_{i,-}(v) - \sum_{j \in \mathcal{N} \setminus i} \varphi_{i|j}(v) \\ \geq (\lambda_i^* - 1) p_{i,-}(v) + p_{i,-}(v) - \sum_{j \in \mathcal{N} \setminus i} \varphi_{i|j}(v) \\ \geq (\lambda_i^* - 1) p_{i,-}(v) - p_i(v) . \tag{36} \end{aligned}$$

We also observe that if $p_{i,-}(v) = 0$, then $-p_i(v) = -\sum_{j \in \mathcal{N} \setminus i} \varphi_{i|j}(v) \leq 0 \leq b_i$ which is a contradiction to (35). Thus,

$$p_{i,-}(v) > 0 . \tag{37}$$

Combining with $-p_i(\boldsymbol{\lambda}^* v) < -p_i(v)$ (35), and (36), we have

$$\lambda_i^* < 1 . \tag{38}$$

From (35), (37), and (38), there exists $\epsilon > 0$ such that

$$\begin{cases} -p_i(\boldsymbol{\lambda}^* v) + \epsilon < b_i < -p_i(v) \\ \lambda_i^* + \epsilon/p_{i,-}(v) < 1 \end{cases} \tag{39}$$

Let $\boldsymbol{\lambda}'$ be defined as follows

$$\begin{cases} \lambda_j' = \lambda_j^* + \epsilon/p_{j,-}(v) & \text{for } j = i \\ \lambda_j' = \lambda_j^* & \text{for } j \neq i . \end{cases} \tag{40}$$

Then $\lambda_i' \in [0, 1]$ for all $i \in \mathcal{N}$ from (39).

Let us consider the payment of party $i$ (i.e., the negative value of the payoff reward)

$$\begin{aligned} -p_i(\boldsymbol{\lambda}' v) &= \lambda_i' p_{i,-}(v) - \sum_{j \in \mathcal{N} \backslash i} \lambda_j' \varphi_{i|j}(v) \\ &= \lambda_i^* p_{i,-}(v) + \epsilon - \sum_{j \in \mathcal{N} \backslash i} \lambda_j^* \varphi_{i|j}(v) \\ &= -p_i(\boldsymbol{\lambda}^* v) + \epsilon \\ &< b_i \end{aligned}$$

where the last inequality is due to (39). Hence, $\boldsymbol{\lambda}'$ satisfies the budget constraint of the party $i$.

Let us consider the payment of party $j \in \mathcal{N} \setminus i$,

$$\begin{aligned} -p_j(\boldsymbol{\lambda}' v) &= \lambda_j' p_{j,-}(v) - \sum_{k \in \mathcal{N} \backslash j} \lambda_k' \varphi_{j|k}(v) \\ &= \lambda_j' p_{j,-}(v) - \sum_{k \in \mathcal{N} \backslash \{i,j\}} \lambda_k' \varphi_{j|k}(v) - \lambda_i' \varphi_{j|i}(v) \\ &= \lambda_j^* p_{j,-}(v) - \sum_{k \in \mathcal{N} \backslash \{i,j\}} \lambda_k^* \varphi_{j|k}(v) \\ &\quad - \left( \lambda_i^* + \frac{\epsilon}{p_{i,-}(v)} \right) \varphi_{j|i}(v) \\ &= \lambda_j^* p_{j,-}(v) - \sum_{k \in \mathcal{N} \backslash j} \lambda_k^* \varphi_{j|k}(v) - \frac{\epsilon}{p_{i,-}(v)} \varphi_{j|i}(v) \\ &= -p_j(\boldsymbol{\lambda}^* v) - \frac{\epsilon}{p_{i,-}(v)} \varphi_{j|i}(v) \\ &\leq b_j - \frac{\epsilon}{p_{i,-}(v)} \varphi_{j|i}(v) \tag{41} \\ &\leq b_j \tag{42} \end{aligned}$$

where (41) is because $\boldsymbol{\lambda}^*$ is the optimal solution to (P) and (42) is because $\varphi_{j|i}(v) \geq 0$ and $p_{i,-}(v) \geq 0$ (from the monotonicity assumption). Hence, $\boldsymbol{\lambda}'$ satisfies the budget constraints of parties in $\mathcal{N} \setminus i$.

As a result, $\boldsymbol{\lambda}'$ is a feasible solution to (P). Moreover, let us consider the objective of the LP in Sec. 4,

$$\sum_{\{k,j\} \subset \mathcal{N}} \lambda_k' \varphi_{j|k}(v) = \sum_{\{k,j\} \subset \mathcal{N}} \lambda_k^* \varphi_{j|k}(v) + \epsilon > \sum_{\{k,j\} \subset \mathcal{N}} \lambda_k^* \varphi_{j|k}(v) \tag{43}$$

which is a contradiction because $\boldsymbol{\lambda}^*$ is the optimal solution to the LP.

Therefore, there does not exist a party $i$ that satisfies (35). It means that if $b_i < -p_i(v)$, then $-p_i(\boldsymbol{\lambda}^* v) = b_i$, i.e., there are not any budget surplus.

## G.3 The Uniqueness of $\lambda$

We will show that given the payoff flows specified by $\lambda_i \varphi_{j|i}(v)$ (from the feasibility & individual rationality properties and the definition of the conditional Shapley value), there exists a unique value of $\boldsymbol{\lambda} = (\lambda_i)_{i=1}^n$ that satisfies the budget constraints and the budget efficiency properties. Hence, this unique value must be the optimal solution $\boldsymbol{\lambda}^*$ to (P).

First, we prove the following lemma.

**Lemma G.3.** *Let $\boldsymbol{\lambda}' \triangleq (\lambda_i')_{i=1}^n$ be the scaling parameter such that the payoff flows specified by $\lambda_i' \varphi_{j|i}(v)$ satisfy the budget constraints and $\lambda_i' \in [0,1]$ for all $\{i,j\} \subset \mathcal{N}$, then $\lambda_i' \leq \lambda_i^*$ where $\boldsymbol{\lambda}^* \triangleq (\lambda_i^*)_{i=1}^n$ is the optimal solution to (P).*

*Proof.* Let us prove by contradiction. If $\exists k \in \mathcal{N}$, $\lambda_k' > \lambda_k^*$, we choose

$$i_* \triangleq \operatorname*{argmax}_{j \in \mathcal{N}}(\lambda_j' - \lambda_j^*) . \tag{44}$$

It follows that

$$\lambda_{i_*}' > \lambda_{i_*}^* . \tag{45}$$

Since $\lambda_{i_*}' \in [0,1]$ (from the constraints of (P)), $\lambda_{i_*}^* \in [0,1)$. It can be seen that if $\lambda_{i_*}^* < 1$, then the budget constraint of party $i_*$ is active (otherwise, we can increase $\lambda_{i_*}^*$ by an $\epsilon > 0$ to obtain a higher value of the objective function while the budget constraints of all parties are still satisfied), i.e.,

$$p_{i_*}(\boldsymbol{\lambda}^* \overrightarrow{v}) = p_{i_*,+}(\boldsymbol{\lambda}^* \overrightarrow{v}) - \lambda_{i_*}^* p_{i_*,-}(v) = \sum_{j \in \mathcal{N} \setminus i_*} \lambda_j^* \varphi_{i_*|j}(v) - \lambda_{i_*}^* \sum_{j \in \mathcal{N} \setminus i_*} \varphi_{j|i_*}(v) = -b_{i_*} . \tag{46}$$

As $\boldsymbol{\lambda}'$ is a feasible solution to (P), the budget constraints are satisfied:

$$p_{i_*,+}(\boldsymbol{\lambda}' \overrightarrow{v}) - \lambda_{i_*}' p_{i_*,-}(v) \geq -b_{i_*} = p_{i_*,+}(\boldsymbol{\lambda}^* \overrightarrow{v}) - \lambda_{i_*}^* p_{i_*,-}(v)$$

where the equality is due to (46). Since $\lambda_{i_*}' > \lambda_{i_*}^*$ (from (45)), $\lambda_{i_*}' p_{i_*,-}(v) > \lambda_{i_*}^* p_{i_*,-}(v)$. Thus,

$$p_{i_*,+}(\boldsymbol{\lambda}' \overrightarrow{v}) - p_{i_*,+}(\boldsymbol{\lambda}^* \overrightarrow{v}) \geq \lambda_{i_*}' p_{i_*,-}(v) - \lambda_{i_*}^* p_{i_*,-}(v) > 0$$

$$\sum_{j \in \mathcal{N} \setminus i_*} (\lambda_j' - \lambda_j^*) p_{ji_*}(v) \geq (\lambda_{i_*}' - \lambda_{i_*}^*) p_{i_*,-}(v) > 0 \tag{47}$$

Let

$$\mathcal{N}_+ \triangleq \{i \in \mathcal{N} | \lambda_i' > \lambda_i^*\}$$

which is not empty due to the assumption (45). Furthermore, since $p_{ij}(v) \geq 0$ for all $\{i,j\} \subset \mathcal{N}$, (47) implies that

$$\sum_{j \in \mathcal{N}_+} (\lambda_j' - \lambda_j^*) p_{ji_*}(v) > (\lambda_{i_*}' - \lambda_{i_*}^*) p_{i_*,-}(v) .$$

Let

$$i' \triangleq \operatorname*{arg\,max}_{j \in \mathcal{N}_+} \lambda_j' - \lambda_j^* .$$

Then,

$$(\lambda_{i'}' - \lambda_{i'}^*) \sum_{j \in \mathcal{N}} p_{ji_*}(v) \geq (\lambda_{i'}' - \lambda_{i'}^*) \sum_{j \in \mathcal{N}_+} p_{ji_*}(v)$$

$$\geq \sum_{j \in \mathcal{N}_+} (\lambda_j' - \lambda_j^*) p_{ji_*}(v)$$

$$> (\lambda_{i_*}' - \lambda_{i_*}^*) p_{i_*,-}(v) .$$

Equivalently,

$$(\lambda'_{i'} - \lambda^*_{i'})p_{i_*,+}(v) > (\lambda'_{i_*} - \lambda^*_{i_*})p_{i_*,-}(v)$$

Furthermore, $p_{i_*,+}(v) \leq p_{i_*,-}(v)$ (otherwise, the reward of party $i$ is positive and the budget constraint of party $i$ is not active which contradicts to (46)), so

$$\lambda'_{i'} - \lambda^*_{i'} > \lambda'_{i_*} - \lambda^*_{i_*} \tag{48}$$

$$\tag{49}$$

which contradicts the definition of $i_*$. $\qquad\square$

Now, let us prove the uniqueness of $\boldsymbol{\lambda}$ such that the payoff flows specified by $p_{ij}(\lambda_i v) = \lambda_i \varphi_{j|i}(v)$ satisfies the budget constraints and the budget efficiency.

*Proof.* From the proof of the existence, the optimal solution $\boldsymbol{\lambda}^*$ to (P) is a value of $\boldsymbol{\lambda}$ which leads to payoff flows that satisfy the budget constraints and the budget efficiency.

To prove by contradiction, we assume that there is another $\boldsymbol{\lambda}'$ that is different from $\boldsymbol{\lambda}^*$ such that the payoff flows specified by $p_{ij}(\lambda'_i v) = \lambda'_i \varphi_{j|i}(v)$ satisfies the budget constraints and the weak efficiency property. Let us define

$$\mathcal{N}_- \triangleq \{i \in \mathcal{N} | \lambda'_i < \lambda^*_i\} \tag{50}$$

$$i_\circ \triangleq \arg\max_{i \in \mathcal{N}_-} \lambda'_i - \lambda^*_i. \tag{51}$$

Then, from Lemma G.3 and $\boldsymbol{\lambda}' \neq \boldsymbol{\lambda}^*$, we have

$$\mathcal{N}_- \neq \emptyset$$
$$\forall i \in \mathcal{N} \setminus \mathcal{N}_-, \quad \lambda'_i = \lambda^*_i. \tag{52}$$

Furthermore, $\lambda^*_{i_\circ} \in [0, 1]$, so $\lambda'_{i_\circ} < 1$. It implies that party $i_\circ$ does not receive the optimal model of value $v(\mathcal{N})$, which implies the followings (since the payoff flows specified by $\boldsymbol{\lambda}^*$ and $\boldsymbol{\lambda}'$ satisfy the budget efficiency property).

$$-p_{i_\circ}(v) > b_{i_\circ}, \quad \text{which implies } p_{i_\circ,-}(v) > p_{i_\circ,+}(v) \tag{53}$$

$$-p_{i_\circ}(\boldsymbol{\lambda}'v) = -p_{i_\circ}(\boldsymbol{\lambda}^*v) = b_{i_\circ}. \tag{54}$$

The latter (54) can be re-written as follows.

$$p_{i_\circ,+}(\boldsymbol{\lambda}'v) - p_{i_\circ,-}(\boldsymbol{\lambda}'v) = p_{i_\circ,+}(\boldsymbol{\lambda}^*v) - p_{i_\circ,-}(\boldsymbol{\lambda}^*v)$$
$$p_{i_\circ,+}(\boldsymbol{\lambda}'v) - p_{i_\circ,+}(\boldsymbol{\lambda}^*v) = p_{i_\circ,-}(\boldsymbol{\lambda}'v) - p_{i_\circ,-}(\boldsymbol{\lambda}^*v)$$
$$\sum_{j \in \mathcal{N} \setminus i_\circ} \lambda'_j p_{ji_\circ}(v) - \sum_{j \in \mathcal{N} \setminus i_\circ} \lambda^*_j p_{ji_\circ}(v) = \lambda'_{i_\circ} p_{i_\circ,-}(v) - \lambda^*_{i_\circ} p_{i_\circ,-}(v)$$
$$\sum_{j \in \mathcal{N} \setminus i_\circ} (\lambda'_j - \lambda^*_j) p_{ji_\circ}(v) = (\lambda'_{i_\circ} - \lambda^*_{i_\circ}) p_{i_\circ,-}(v). \tag{55}$$

However, from (51), (52), and (53),

$$\sum_{j \in \mathcal{N} \setminus i_\circ} (\lambda'_j - \lambda^*_j) p_{ji_\circ}(v) < (\lambda'_{i_\circ} - \lambda^*_{i_\circ}) \sum_{j \in \mathcal{N} \setminus i_\circ} p_{ji_\circ}(v)$$
$$= (\lambda'_{i_\circ} - \lambda^*_{i_\circ}) p_{i_\circ,+}(v)$$
$$< (\lambda'_{i_\circ} - \lambda^*_{i_\circ}) p_{i_\circ,-}(v)$$

which contradicts (55). Thus, $\boldsymbol{\lambda}'$ must be the same as $\boldsymbol{\lambda}^*$. $\qquad\square$

It is noted that $\lambda^*_i \varphi_{j|i}(v) = \varphi_{j|i}(\lambda^*v)$ due to the linearity property of the conditional Shapley value. Thus, the payoff flows $\lambda^* \varphi_{j|i}(v)$ for all $\{i, j\} \subset \mathcal{N}$ are the conditional Shapley values of the adjusted value function $\overrightarrow{v'} = \boldsymbol{\lambda}^* \overrightarrow{v} = (\lambda^*_i v)^n_{i=1}$ where $\boldsymbol{\lambda}^*$ is the optimal solution to (P). Therefore, this concludes the proof of Lemma 4.1.

# H   Proof of Weak Efficiency

In this section, we prove that the adjusted payoff flows also satisfy the weak efficiency property defined in Sec. 4: There always exists a party that receives the optimal trained on the aggregated dataset $\mathcal{D}_\mathcal{N}$ (i.e., the model of value $v(\mathcal{N})$) in the following section.

As discussed in Sec. 3.2, for all $i \in \mathcal{N}$, the performance of the model obtained by party $i$ is

$$v(i) + p_{i,-}(\lambda_i^* v) = v(i) + \lambda_i^* \sum_{j \in \mathcal{N} \backslash i} \varphi_{j|i}(v) . \tag{56}$$

To prove the weak efficiency of the above allocation scheme, we need to show there exists a party $i \in \mathcal{N}$ such that $\lambda_i^* = 1$, i.e., party $i$ receives a model of the value

$$
\begin{aligned}
v(i) + \lambda_i^* \sum_{j \in \mathcal{N} \backslash i} \varphi_{j|i}(v) &= v(i) + \sum_{j \in \mathcal{N} \backslash i} \varphi_{j|i}(v) \\
&= v(i) + (v(\mathcal{N} - v(i)) \text{ (due to the fair cost property)} \\
&= v(\mathcal{N}) .
\end{aligned}
$$

In other words, party $i$ receives the optimal model trained on all of the aggregated dataset $\mathcal{D}_\mathcal{N}$. Hence, we prove the weak efficiency of the payoff flows specified by $p_{ij}(\overrightarrow{v}') = \lambda_i^* \varphi_{j|i}(v)$ by proving the following lemma.

**Lemma H.1.** *There always exists $i \in \mathcal{N}$ such that $\lambda_i^* = 1$ where $\boldsymbol{\lambda}^* \triangleq (\lambda_i^*)_{i=1}^n$ is the optimal solution to* (P).

*Proof.* **Case 1:** $\exists i \in \mathcal{N}$, $b_i > 0$. In such a case, $\lambda_i^* > 0$ (otherwise, we can increase $\lambda_i^*$ by a small positive value and obtain a solution to (P) that is better than $\boldsymbol{\lambda}^*$, which is a contradiction). Hence,

$$\max_{j \in \mathcal{N}} \lambda_j^* \geq \lambda_i^* > 0 . \tag{57}$$

**Case 1.1:** $\forall j \in \mathcal{N}$, $p_j(\boldsymbol{\lambda}^* \overrightarrow{v}) = 0$. Let $m \triangleq \max_{j \in \mathcal{N}} \lambda_j^*$, then $m > 0$ from (57). Furthermore, since $\boldsymbol{\lambda}^*$ is the optimal solution to (P) (it satisfies the constraint $\lambda_j \in [0, 1]$), so $m \in (0, 1]$.

Suppose that $m \in (0, 1)$. We can define $\boldsymbol{\lambda}' = \frac{1}{m} \boldsymbol{\lambda}^*$, then we can show that $\boldsymbol{\lambda}'$ is a solution to (P) that is better than the optimal $\boldsymbol{\lambda}^*$, which is a contradiction.

Recall the setting of case 1.1: $\forall j \in \mathcal{N}$, $p_j(\boldsymbol{\lambda}^* \overrightarrow{v}) = 0$. Hence, $-p_j(\boldsymbol{\lambda}' \overrightarrow{v}) = -\frac{1}{m} p_j(\boldsymbol{\lambda}^* \overrightarrow{v}) = 0 \leq b_j$ (from the linearity property of the conditional Shapley value) and $\frac{1}{m} \lambda_j^* = \frac{\lambda_j^*}{\max_{k \in \mathcal{N}} \lambda_k^*} \in [0, 1]$ (the definition of $m$), which means that $\boldsymbol{\lambda}'$ is a feasible solution to (P). Furthermore, since $m \in (0, 1)$,

$$\sum_{\{j,k\} \subset \mathcal{N}} \frac{1}{m} \lambda_j^* \varphi_{k|j}(v) > \sum_{\{j,k\} \subset \mathcal{N}} \lambda_j^* \varphi_{k|j}(v) .$$

Therefore, $\boldsymbol{\lambda}'$ is a solution to (P) that is better than the optimal $\boldsymbol{\lambda}^*$, which is a contradiction. Hence, $m \notin (0, 1)$. It means that $m = 1$, i.e., by letting $i = \arg\max_{k \in \mathcal{N}} \lambda_k^*$, then $\lambda_i^* = 1$.

**Case 1.2:** $\exists j \in \mathcal{N}$, $p_j(\boldsymbol{\lambda}^* \overrightarrow{v}) \neq 0$.

From the balance property in Sec. 3.1,

$$\sum_{i \in \mathcal{N}} p_i(\boldsymbol{\lambda}^* \overrightarrow{v}) = 0 .$$

Therefore, if there exists $j \in \mathcal{N}$ such that $p_j(\boldsymbol{\lambda}^* \overrightarrow{v}) \neq 0$, then there exists $i \in \mathcal{N}$ such that $p_i(\boldsymbol{\lambda}^* \overrightarrow{v}) > 0$ for the above balance property to hold. We will show that $\lambda_i^* = 1$. To prove by contradiction, suppose $\lambda_i^* < 1$, since

$$p_i(\boldsymbol{\lambda}^* \overrightarrow{v}) > 0$$

equivalently,

$$\sum_{j \in \mathcal{N} \backslash i} \lambda_j \varphi_{i|j}(v) - \lambda_i \sum_{j \in \mathcal{N} \backslash i} \varphi_{j|i}(v) > 0 .$$

Since $\lambda_i^* < 1$, there exists $\epsilon > 0$ such that

$$\begin{cases} \lambda_i' = \lambda_i^* + \epsilon \leq 1 \\ \sum_{j \in \mathcal{N} \setminus i} \lambda_j \varphi_{i|j}(v) - \lambda_i' \sum_{j \in \mathcal{N} \setminus i} \varphi_{j|i}(v) > 0 \,. \end{cases} \tag{58}$$

Let $\boldsymbol{\lambda}' = (\lambda_j')_{j=1}^n$ be defined as follows

$$\begin{cases} \lambda_i' = \lambda_i^* + \epsilon \\ \lambda_j' = \lambda_j^* \qquad \text{for } j \in \mathcal{N} \setminus i \,. \end{cases}$$

From (58), it follows that $\lambda_j' \in [0,1]$ for all $j \in \mathcal{N}$. Furthermore, $p_i(\boldsymbol{\lambda}^* \overrightarrow{v}) > 0$ (the setting of the case 1.2) so $-p_i(\boldsymbol{\lambda}' \overrightarrow{v}) < 0 \leq b_i$. Hence, for $\boldsymbol{\lambda}'$ to be a feasible solution to (P), we need to show that

$$-p_j(\boldsymbol{\lambda}' \overrightarrow{v}) \leq b_j \ \forall j \in \mathcal{N} \setminus i \,.$$

In fact, for all $j \in \mathcal{N} \setminus i$,

$$\begin{aligned} p_j(\boldsymbol{\lambda}' \overrightarrow{v}) &= \sum_{k \in \mathcal{N} \setminus j} \lambda_k' \varphi_{j|k}(v) - \lambda_j' \sum_{k \in \mathcal{N} \setminus j} \varphi_{k|j}(v) \\ &= \lambda_i' \varphi_{j|i}(v) + \sum_{k \in \mathcal{N} \setminus \{i,j\}} \lambda_k' \varphi_{j|k}(v) - \lambda_j' \sum_{k \in \mathcal{N} \setminus j} \varphi_{k|j}(v) \\ &= \epsilon \varphi_{j|i}(v) + \sum_{k \in \mathcal{N} \setminus j} \lambda_k^* \varphi_{j|k}(v) - \lambda_j^* \sum_{k \in \mathcal{N} \setminus j} \varphi_{k|j}(v) \\ &= \epsilon \varphi_{j|i}(v) + p_j(\boldsymbol{\lambda}^* \overrightarrow{v}) \\ &\geq p_j(\boldsymbol{\lambda}^* \overrightarrow{v}) \\ &\geq -b_j \,. \end{aligned}$$

Hence, $-p_j(\boldsymbol{\lambda}' \overrightarrow{v}) \leq b_j$. Therefore, $\boldsymbol{\lambda}'$ is a feasible solution to the LP in Sec. 4. Furthermore,

$$\sum_{\{j,k\} \subset \mathcal{N}} \lambda_j' \varphi_{k|j}(v) = \left( \sum_{j \in \mathcal{N} \setminus i} \epsilon \varphi_{j|i}(v) \right) + \left( \sum_{\{j,k\} \subset \mathcal{N}} \lambda_j^* \varphi_{k|j}(v) \right) > \sum_{\{j,k\} \subset \mathcal{N}} \lambda_j^* \varphi_{k|j}(v) \tag{59}$$

which is a contradiction since $\boldsymbol{\lambda}^*$ is the optimal solution to (P). Hence $\lambda_i^* = 1$.

**Case 2: $\boldsymbol{b} = \boldsymbol{0}$.** It is noted that (P) is always feasible (solution $\boldsymbol{\lambda} = \boldsymbol{0}$) and bounded (bounded by $\sum_{\{i,j\} \subset \mathcal{N}} \varphi_{j|i}(v)$) for all $\mathbf{b} \geq \boldsymbol{0}$. Thus, the objective function of the LP is continuous in $\mathbf{b}$. As the above shows that $\forall \mathbf{b} \geq \boldsymbol{0}$ such that there exists $i \in \mathcal{N}$, $b_i > 0$, there exists $j \in \mathcal{N}$ such that $\lambda_j^* = 1$. The continuity implies that at $\mathbf{b} = \boldsymbol{0}$, there exists $j \in \mathcal{N}$ such that $\lambda_j^* = 1$. □

# I   Extension to Performance Constraints

Suppose for all $i \in \mathcal{N}$, party $i$ has a budget $b_i$ and a *desirable model performance* $r_i$. Then, apart from the budget constraint that prevents party $i$ from expending an amount larger than $b_i$, we introduce the *performance constraint* which states that party $i$ is not interested in obtaining a model with a performance larger than $r_i$.

Given $v$, the budget $\boldsymbol{b} \triangleq (b_i)_{i=1}^n$, and the desirable performance $\boldsymbol{r} \triangleq (r_i)_{i=1}^n$, we would like to find the adjusted value function $\overrightarrow{v}' \triangleq (v_i')_{i=1}^n$ such that the payoff flows $p_{ij}(v_i') = \varphi_{j|i}(v_i')$ meet not only the budget constraints, i.e., $-p_i(\overrightarrow{v}') \leq b_i$, but also the performance constraints, i.e.,

$$v_i'(\mathcal{N}) \leq \max(v(i), r_i) \,. \tag{60}$$

By enforcing the feasibility & individual rationality, the marginal contribution consistency, and the following performance-budget efficiency, the adjusted value functions are uniquely defined.

**Performance-budget efficiency.** Assuming $r_i = \infty$ if party $i$ does not have a performance constraint, the performance-budget efficiency encourages every party to obtain the *budget-constrained desirable model* that it can afford (i.e., the model with the value closest to but not exceeding the *desirable model value* $\min(v(\mathcal{N}), r_i)$ due to the budget constraint) by expending as much of its budget as possible, i.e., for all $i \in \mathcal{N}$,

- If party $i$ can afford the *desirable model value* $\min(v(\mathcal{N}), r_i)$, then it receives the model of value $\min(v(\mathcal{N}), r_i)$ under the adjusted payoff flows.

- If party $i$ cannot afford the *desirable model value* $\min(v(\mathcal{N}), r_i)$ due to its budget constraint, then the adjusted payoff is exactly $-b_i$, i.e., $-p_i(\overrightarrow{v}') = b_i$, i.e., the budget is fully consumed.

**Lemma I.1.** *Let $\boldsymbol{\lambda}^* \triangleq (\lambda_i^*)_{i=1}^n$ be the solution to the following linear programming problem:*

$$
\begin{aligned}
\textit{maximize} \quad & \sum_{\{i,j\} \subset \mathcal{N}} \lambda_i \varphi_{j|i}(v) \\
\textit{subject to} \quad & \lambda_i \sum_{j \in \mathcal{N} \setminus i} \varphi_{j|i}(v) - \sum_{k \in \mathcal{N} \setminus i} \lambda_k \varphi_{i|k}(v) \leq b_i, \quad \forall i \in \mathcal{N} \\
& v(i) + \lambda_i p_{i,-}(v) \leq \max(v(i), r_i), \quad \forall i \in \mathcal{N} \\
& \lambda_i \in [0,1], \quad \forall i \in \mathcal{N} .
\end{aligned} \tag{Q}
$$

*The feasibility & individual rationality, the marginal contribution consistency, the performance-budget efficiency, and the constraints uniquely determine the following adjusted value functions*

$$ \forall i \in \mathcal{N}, \ \forall \mathcal{C} \subset \mathcal{N} \setminus i, \ v_i'(\mathcal{C} \cup i) = \lambda_i^* v(\mathcal{C} \cup i) + (1 - \lambda_i^*) v(i) . $$

*Then, from the fairness properties in Sec. 3.2, the payoff flow $p_{ij}(v_i')$ is uniquely defined as the conditional Shapley value $\varphi_{j|i}(v_i')$ which can be directly computed from $\varphi_{j|i}(v)$:*

$$ \forall j \in \mathcal{N} \setminus i, \ p_{ij}(v_i') = \varphi_{j|i}(v_i') = \varphi_{j|i}(\lambda_i^* v) = \lambda_i^* \varphi_{j|i}(v) . $$

It is noted that the first set of constraints in (Q) are the budget constraints while the second set of constraints are the performance constraints. The proof of Lemma I.1 can be obtained in a similar fashion that of Lemma 4.1 in App. G by observing that the LP (Q) is equivalent to

$$
\begin{aligned}
\textit{maximize} \quad & \sum_{\{i,j\} \subset \mathcal{N}} \lambda_i \varphi_{j|i}(v) \\
\textit{subject to} \quad & \lambda_i \sum_{j \in \mathcal{N} \setminus i} \varphi_{j|i}(v) - \sum_{k \in \mathcal{N} \setminus i} \lambda_k \varphi_{i|k}(v) \leq b_i, \quad \forall i \in \mathcal{N} \\
& \lambda_i \in \left[0, \min\left(1, \frac{\max(v(i), r_i) - v(i)}{p_{i,-}(v)}\right)\right], \quad \forall i \in \mathcal{N} .
\end{aligned} \tag{61}
$$

Furthermore, we can follow the proof of the weak efficiency in App. H to show the following variant of the weak efficiency property.

**A variant of weak efficiency.** There always exists a party in $\mathcal{N}$ that receives the model with the *desirable model value* $\min(v(\mathcal{N}), r_i)$, i.e., the optimal model trained on $\mathcal{D}_{\mathcal{N}}$ (of value $v(\mathcal{N})$) or the desirable model of value $r_i$. It is noted that the desirable model value of a party only depends on the performance constraint and does not depend on the budget constraint. In other words, this property implies that there always exists a party that is not constrained by its budget. For example, in the extreme case when all parties have zero budgets, i.e., $\mathbf{b} = \mathbf{0}$, there still exists a party that receives the model reward as if it has an unlimited budget.

# J  Additional Experiments

In all experiments, we set $\nu = 1$ for simplicity. The experiments are performed on a machine with Intel i7-9750H and 16GB of RAM.

## J.1  MNIST

The MNIST dataset [10] contains $70,000$ $28 \times 28$ gray-scale images of handwritten digits, i.e., the label of an image is one of the 10 digits. A party, denoted as $[s\text{-}e]$ ($s \leq e$), owns a subset of the MNIST training dataset (of size $60,000$) that are labeled with digits $s, s+1, \ldots, e$. There are 5 parties in $\mathcal{N}$: $[0]$ (i.e., [0-0]), [1-2], [0-3], [3-5], and [6-9]. The model performance is measured by the prediction accuracy on a separate dataset of size $10,000$.

The ML model is a neural network with 2 hidden layers each of which consists of $64$ neurons with the ReLU activation functions. The output layer is a softmax layer that consists of $10$ neurons. The parameters of the network are trained with the Adam optimizer. The learning rate is set to 0.001. The batch size is set to $64$. The number of epochs is $5$.

## J.2 CIFAR-10

The CIFAR-10 dataset [9] contains $60,000$ $32 \times 32$ color images. These images are labeled as one of the 10 classes: 0 (airplane), 1 (automobile), 2 (bird), 3 (cat), 4 (deer), 5 (dog), 6 (frog), 7 (horse), 8 (ship), and 9 (truck), each of which consists of 6000 images. The training dataset of $50,000$ images is distributed to 4 parties. Parties 1, 2, 3, and 4 own data of labels $[1, 8, 9]$, $[0, 2, 3]$, $[4, 5]$, and $[6, 7]$, respectively. The model performance is measured by the prediction accuracy on a separate dataset of size $10,000$.

The ML model is a neural network with 3 convolutional layers with the number of filters are 32, 64, and 64. The kernel size of these layers is set to $(3, 3)$. There are 2 max-pooling layers with the pool size $(2, 2)$ after the first 2 convolutional layers. The output of the third convolutional layer is flattened and passed to a layer with 64 neurons and linear activation functions. The output layer is a softmax layer that consists of 10 neurons. The parameters of the network are trained with the Adam optimizer. The learning rate is set to $0.001$. The batch size is set to $512$. The number of epochs is 20.

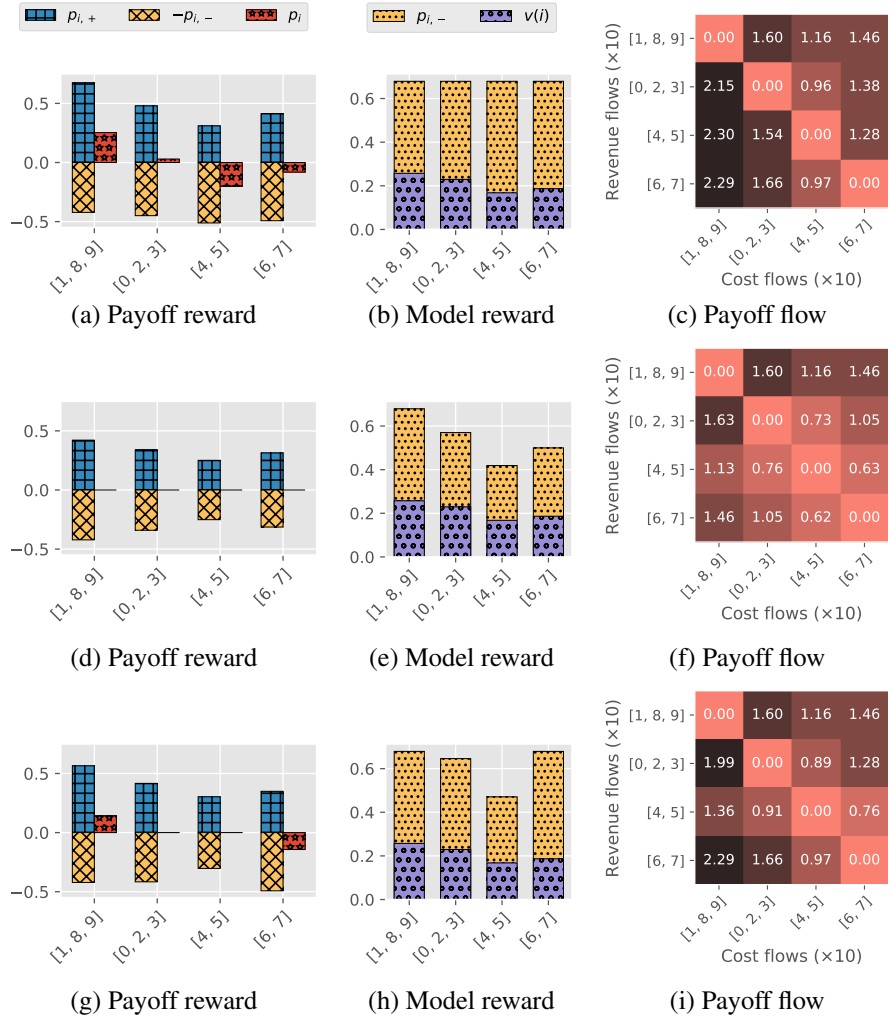

Figure 3: Plots of the payoff, model rewards, and payoff flows in the CIFAR-10 experiments with budget constraints: (a-c) $\mathbf{b} = \infty \mathbf{1}$, (d-f) $\mathbf{b} = \mathbf{0}$, and (g-i) $b_4 = \infty$ and $b_i = 0$ for $i \neq 4$.

Figs. 3a and 3b show the payoff and model rewards when there are not any budget constraints (i.e., $\mathbf{b} = \infty \mathbf{1}$), respectively. In Fig. 3a, parties 3 ($[4, 5]$) and 4 ($[6, 7]$) with smaller datasets (in comparison to the other parties) receive negative payoff rewards. Among parties 1 and 2, we observe that the dataset of party 1 is more valuable as it receives a higher payoff reward. In Fig. 3b, all parties receive models with the same performance of $v(\mathcal{N})$ since there are not any budget

constraints. While the work of [13] can address the case of no budget constraints, it results in a payoff vector $[0.15, 0.01, -0.12, -0.04]$ which does not satisfy the linear value property in Sec. 3.1 and the linear property.

We also note that the work of [13] distributes the total cost of a party to all parties including itself. In contrast, we only distribute the total cost of a party to other parties because they are the parties that cause the increase in the model performance of the party.

Figs. 3d and 3e show the payoff and model rewards when all parties are not allowed to spend any payoff (i.e, $\mathbf{b} = \mathbf{0}$), respectively. Due to no budget, the payoff rewards are $0$ for all parties (see Fig. 3d). However, in Fig. 3e, we observe that they still benefit from the collaboration by obtaining better models (of value $v(i) + p_{i,-}$) than their own models (of value $v(i)$). In particular, party $1$ receives the optimal model trained on the aggregated dataset $\mathcal{D}_{\mathcal{N}}$ (of value $v(\mathcal{N})$), which is consistent with the weak efficiency property. Thus, when monetary payoffs are not allowed, our allocation scheme is still able to motivate parties to fairly collaborate using their data. While the $\rho$-Shapley value [16] can address the problem of allocating the model reward in this scenario (it forbids the use of monetary payoffs), the model reward allocation in Fig. 3d cannot be obtained from the $\rho$-Shapley value (by varying its parameters). When $\rho < 1$, the $\rho$-Shapley value does not satisfy the linear property, while our proposed allocation scheme does.

Figs. 3g and 3h show the payoff and model rewards when party $4$ ($[6, 7]$) has an unlimited budget and other parties have budgets of $0$, i.e., $b_4 = \infty$ and $b_i = 0$ for $i \neq 4$. While it is intuitive that with an unlimited budget, party $4$ can obtain the model trained on $\mathcal{D}_{\mathcal{N}}$ (of value $v(\mathcal{N})$), it is less obvious that parties $2$ and $3$ also have the performance of their models improved in comparison to the previous case (by comparing Fig. 3f and Fig. 3d). The reason is that revenues of parties $2$ and $3$ from party $4$ increase (comparing the last rows in Figs. 3f and 3i). These increased revenues are used to further improve the models of $2$ and $3$, e.g., by getting more contributions from party $1$ (comparing the first columns in Figs. 3f and 3i). Party $1$ can obtain an additional payoff reward for its contributions to other parties in Fig. 3g. Thus, according to our allocation scheme, increasing the budget of a party benefits not only itself but also other parties.

Another observation in Fig. 3g is that party $4$ makes a larger payment in comparison to Fig. 3a where all parties do not have any budget constraints (by comparing the red star bars of party $4$ in the $2$ figures). This is because in Fig. 3a, party $4$ can obtain more revenue from parties $2$ and $3$, while in Fig. 3e, party $4$ cannot since both parties $2$ and $3$ have budget constraints of $b_2 = b_3 = 0$. We can also observe it by comparing the last columns in Figs. 3c and 3i: the revenue flows from parties $2$ and $3$ to party $4$ reduces when the budget constraints are imposed on parties $2$ and $3$.

### J.3 IMDB Movie Reviews Dataset

The IMDB movie reviews dataset [11] contains $50,000$ movie text reviews. Each review is labeled as either positive or negative. We randomly distribute $20,000$ reviews to $5$ parties equally. The model performance is measured by the prediction accuracy on a separate dataset of size $25,000$.

Movie reviews are encoded (via standardization, tokenization, and vectorization with the text vectorization layer in Tensorflow) as positive integer vectors of fixed length $250$. Then, these positive integers are transformed into dense vectors of size $16$ using the embedding layer in Tensorflow. Next, we average over the sequence dimension using a global average pooling layer. The output vectors of size $16$ are piped through a hidden layer with $16$ neurons and linear activation functions. Last, the output layer is a layer with $1$ neuron and the sigmoid activation function. We also apply drop-out layers with the drop-out rate of $0.2$ after the embedding layer and the global average pooling layer. The parameters of the network are trained with the Adam optimizer. The learning rate is set to $0.001$. The batch size is set to $32$. The number of epochs is $10$.

Figs. 4a and 4b show the payoff and model rewards when there are not any budget constraints (i.e., $\mathbf{b} = \infty\mathbf{1}$), respectively. While we randomly distribute the data to all parties, we observe that party $1$ has the most valuable dataset as it receives the highest payoff reward in Fig. 4a and $v(1)$ is largest among $v(i)$ for $i \in \mathcal{N}$ in Fig. 4b. In Fig. 4b, all parties receive models with the highest performance as there are not any budget constraints. While the work of [13] can address the case of no budget constraints, it results in a payoff vector $[0.042, -0.002, -0.020, -0.017, -0.002]$ which does not satisfy the linear value property in Sec. 3.1 and the linearity property. We also note that the work of [13] distributes the total cost of a party to all parties including itself. In contrast, we only distribute

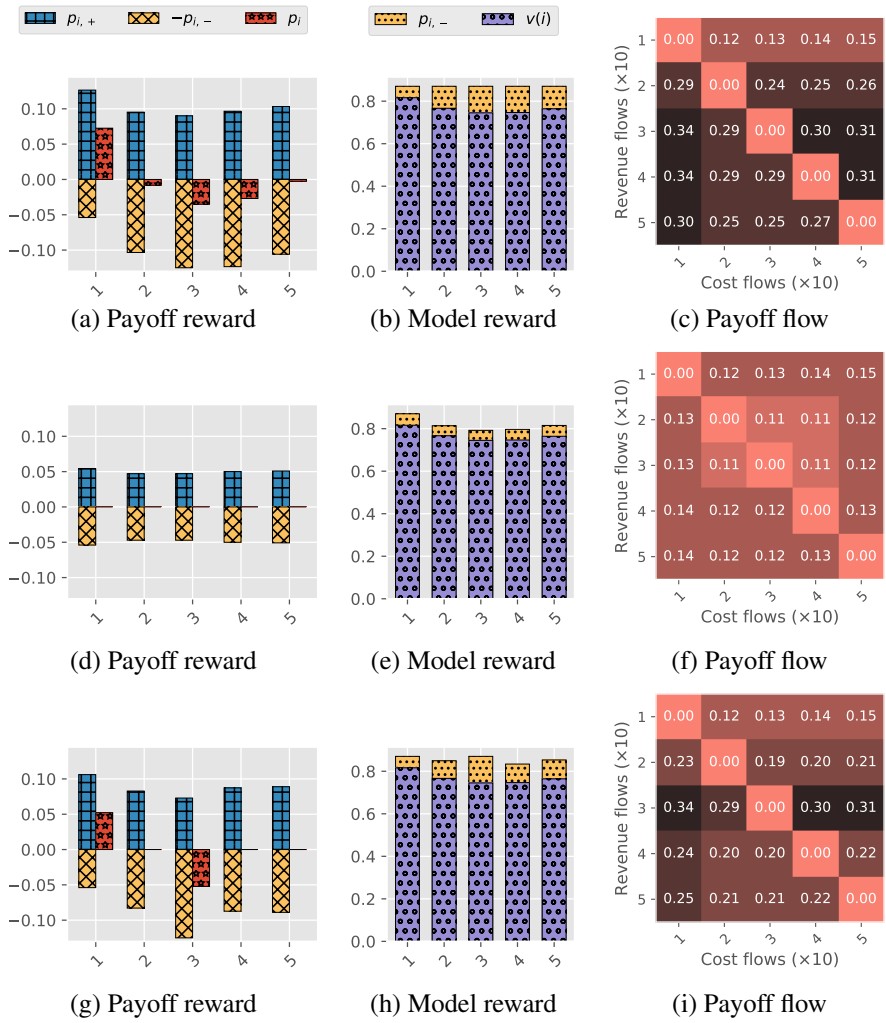

Figure 4: Plots of the payoff, model rewards, and payoff flows in the sentiment analysis of the IMDB movie reviews dataset with budget constraints: (a-c) $\mathbf{b} = \infty\mathbf{1}$, (d-f) $\mathbf{b} = \mathbf{0}$, and (g-i) $b_3 = \infty$ and $b_i = 0$ for $i \neq 3$.

the total cost of a party to other parties because they are the parties that cause the increase in the model performance of the party.

Figs. 4d and 4e show the payoff and model rewards when all parties are not allowed to spend any payoff (i.e, $\mathbf{b} = \mathbf{0}$), respectively. Due to no budget, the payoff rewards are 0 for all parties (see Fig. 4d). However, in Fig. 4e, they still benefit from the collaboration by obtaining better models (of value $v(i) + p_{i,-}$) than their own models (of value $v(i)$). In particular, we observe that party 1 receives the optimal model trained on the aggregated dataset $\mathcal{D}_{\mathcal{N}}$ (of value $v(\mathcal{N})$), which is consistent with the weak efficiency property. Thus, when monetary payoffs are not allowed, our allocation scheme is still able to motivate parties to fairly collaborate using their data. While the $\rho$-Shapley value [16] can address the problem of allocating the model reward in this scenario (it does not allow monetary payoffs), the model reward allocation in Fig. 4e cannot be obtained from the $\rho$-Shapley value (by varying its parameters). When $\rho < 1$, the $\rho$-Shapley value does not satisfy the linearity property, while our proposed allocation scheme does.

Figs. 4g and 4h show the payoff and model rewards when party 3 has an unlimited budget and other parties have budgets of 0, i.e., $b_3 = \infty$ and $b_i = 0$ for $i \neq 3$. While it is intuitive that with an unlimited budget, party 3 can obtain the optimal model trained on $\mathcal{D}_{\mathcal{N}}$ (of value $v(\mathcal{N})$, it is less obvious that parties 2, 4, and 5 also have the performance of their models improved in comparison

to the previous case (by comparing Fig. 4h and Fig. 4e). The reason is that revenues of parties 2, 4, and 5 from party 3 increase (comparing the last rows in Fig. 4f and 4i). These increased revenues are used to further improve the models of 2, 4, and 5, e.g., by getting more contributions from party 1 (comparing the first columns in Figs. 4f and 4i). Party 1 can obtain an additional payoff reward for its contributions to other parties in Fig. 4g. Thus, according to our allocation scheme, increasing the budget of a party benefits not only itself but also other parties.

Similar to the previous CIFAR-10 experiment, another observation in Fig. 4g is that party 3 makes a larger payment in comparison to Fig. 4a where all parties do not have any budget constraints (by comparing the red star bars of party 3 in the 2 figures). This is because in Fig. 4a, party 3 can obtain more revenue from parties 2, 4, and 5, while in Fig. 4e, party 4 cannot since parties 2, 4, and 5 have budget constraints of $b_2 = b_4 = b_5 = 0$. We can also observe it by comparing the 3-rd columns in Figs. 4f and 4i: the revenue flows from parties 2, 4, and 5 to party 3 reduces when the budget constraints are imposed on parties 2, 4, and 5.