# OpenReview forum: "Trade-off between Payoff and Model Rewards in Shapley-Fair Collaborative Machine Learning"
_NeurIPS.cc/2022/Conference — NeurIPS 2022 Accept_

### Official Review · Reviewer_nJsn · 2022-07-01

**Rating:** 5
**Confidence:** 2
**Soundness:** 3 good
**Presentation:** 3 good
**Contribution:** 3 good

**Summary:**

The paper studies collaborative machine learning, where data from multiple parties is aggregated to train a machine learning model with better performance. It is concerned with fairness in the sense that parties should receive a reward that accounts for how much they contributed to the quality of the model, as measured by the Shapley value. The paper notably presents a set of desirable properties of the payoff flows between parties from which it derives a fair allocation scheme. This approach allows to address settings with or without budget constraints. The method is evaluated on a simulation with segments from the MNIST dataset.

**Questions:**

* What is the difference between this work and [Sim et al., 2020] ([14]), in the use of Shapley values to measure the contribution of a party?

* Sec. 4: why should we consider fair to reduce the performance of the model a party receive, in the presence of budget constraints?


**Limitations:**

The limitations and societal impact are not discussed in this work. In particular, the authors did not discuss the privacy issues involved when different parties share their data.

**Strengths And Weaknesses:**

Strengths: The paper is overall well written. The fairness properties of payoff flows for fair allocation are interesting and allow to address both budgeted and non-budgeted settings. The scheme based on payoff rewards is limited to non-constrained problems but it is simpler to understand. The presentation of both types of allocation schemes allows to understand well the advantages and drawbacks of either approaches.

Weaknesses: The contribution of the paper is not very clear: How do the properties defined in the paper differ from classical axioms of cooperative game theory? This is important because the paper is not technical, and its main interest thus lies in the conceptual framework studied.

Typos: L60 et -> It, L388 collaboration -> collaborative

---

> ### Author Response · Authors · 2022-08-02
> **Response to Reviewer nJsn (part 1)**
>
> We would like to thank you for acknowledging our well-written paper and the interesting fairness properties that we proposed.
>
> ---
>
> We would like to clarify the major difference between our work and [14] and the reason behind choosing to reduce the performance of the model a party receives.
>
> > 1. What is the difference between this work and [Sim et al., 2020] ([14]), in the use of Shapley values to measure the contribution of a party?
>
> Our work is different from [14] in both aspects: the problem and the solution, which are described in lines 50-64.
> * Regarding the problem, the problem setting in [14] concerns only the model reward, which is a special case of our problem setting (whereas we can trade off between payoff and model rewards), as mentioned in lines 58-61.
> * Regarding the solution,
>   * Approach:
>     * We follow the axiomatic approach in which we identify a set of fairness properties that uniquely define an allocation scheme. The work of [14] only proposes a modification of the Shapley value without showing its uniqueness. The importance of uniqueness is shown in lines 52-57. In particular, it is unclear in the work of [14] which modification (e.g., which value of $\rho$ in $\rho$-Shapley value) is preferable.
>     * Furthermore, while our modified allocation scheme is the result of the adjustment in the value function, the work of [14] directly alters the allocation scheme without considering the contributions of a party to another after the modification. Hence, the modified Shapley value in [14] no longer reflects the contribution properly.
>
>   * Property: The solution in [14] does not satisfy the linearity property (a fairness property) while ours does (lines 64).  It also implies that even in the special case of our problem setting where the solution of [14] is applicable, the solution of [14] is not the same as our solution (lines 61-63).
>
>
> The difference is also mentioned in lines 371-374 in the experiments section.
>
>
> > 2. Sec. 4: why should we consider fair to reduce the performance of the model a party receive, in the presence of budget constraints?
>
> For simplicity, let us assume party A is a dummy party (i.e., having no contribution to other parties).
> Suppose that using the solution in Section 3.1, party A can obtain an increase of 70% in the model performance from the other parties and A pays 100 USD to the other parties to maintain fairness. In other words, the payment of 100 USD is for the increase of 70% in the model performance. Now, if party A can only afford to expend 50 USD, we think it is reasonable that the increase in the model performance of A should be reduced (to be less than 70%) to be fair.
>
> We would like to note that even though the performance of the model a party receives is reduced, it is still at least the performance of the model that the party can achieve on its own due to the individual rationality property (Equation 8). In other words, it guarantees that the model performance of a party is not reduced by joining the collaboration.
>
> If we view the problem in [14] as a special case of ours where the budget is $0$ for all parties, then the solution in [14] also implies that the performance of the model a party receives is reduced.

---

> > ### Author Response · Authors · 2022-08-02
> > **Response to Reviewer nJsn (part 2)**
> >
> > We would like to address your other concerns below.
> >
> > > The contribution of the paper is not very clear: How do the properties defined in the paper differ from classical axioms of cooperative game theory? This is important because the paper is not technical, and its main interest thus lies in the conceptual framework studied.
> >
> > Our work is different from classical axioms of cooperative game theory as we explained in lines 112-118 that the Shapley value cannot be directly used. Remark 3.3 further emphasizes that the same training dataset can be used to train $n$ models for $n$ parties, so data is different from the usual commodity where it is often sold to one buyer. Furthermore, to the best of our knowledge, a solution to trade off between payoff and model rewards has not been studied in the traditional cooperative game theory.
> >
> > While introducing desirable properties for fairness seems to be conceptual, these properties are chosen not only by the intuition but also to uniquely define the allocation scheme (with and without budget constraints). The latter is a significant technical challenge from our perspective.
> >
> > > Typos: L60 et -> It, L388 collaboration -> collaborative
> >
> > Thank you for spotting the typos. We will correct them in our revised paper.
> >
> > > The limitations and societal impact are not discussed in this work. In particular, the authors did not discuss the privacy issues involved when different parties share their data.
> >
> > From our perspective, our work does not have any significant societal impact, as shown in our Checklist. If you have any concerns regarding the societal impact of this work, please let us know. We will update the paper accordingly.
> >
> > In this work, we also assume that the parties are truthful in contributing the data. We will clarify this assumption in our revised paper.
> >
> > ---
> >
> > We sincerely hope that these clarifications and the two experiments in Appendices H.2 (CIFAR-10) and H.3 (IMDB) (that you may have missed) will improve your opinion of our work.

---

### Official Review · Reviewer_xYwo · 2022-07-10

**Rating:** 7
**Confidence:** 3
**Soundness:** 4 excellent
**Presentation:** 3 good
**Contribution:** 4 excellent

**Summary:**

The authors considered an interesting and practical problem where groups of data owner contribute their data to collaborative build an ML model that is better than models they build on their own. The key contribution comes into designing a unique solution concept that simultaneously consider how monetary payoffs and model rewards should be assigned to each contributor using (Conditional) Shapley values.

**Questions:**

On the contrary, if a party has a constraint or minimal requirement on model performance instead of monetary budget, is your proposed optimisation framework applicable in this case? More concretely, if a party only requires the model to be as good as 90% of the best performing model in order to obtain some discounted payoffs, is your method applicable in this situation?

Another question: is the monotonicity assumption of s needed in your formulation? It seems to be needed to ensure joining the cooperation will be better than not joining, but in terms of Shapley values computation there is no such restriction needed when defining a value function? Can you comment a bit on this please?

**Strengths And Weaknesses:**

Strengths:
- proposing a solution that handles monetary payoff + model rewards simultaneously is not something the literature has considered before.
- Communication of ideas are mostly clear besides the use of payoff flows (see below).

Weaknesses:
- The introduction of payoff flow and conditional SVs weren't apparent to readers until getting to Sec.4, and it seemed to complicate the payoff allocation scheme. It was only made clear later in section 4 that we can handle the budget constraints by modifying the value functions of all parties in the Conditional SV. Maybe this intuition should be made a bit earlier to make it easier to follow.

---

> ### Author Response · Authors · 2022-08-02
> **Response to Reviewer xYwo**
>
> We would like to thank you for recognizing our clear writing and significant contributions, and providing useful feedback which we will incorporate into our revised paper.
>
> ---
>
> We would like to discuss your interesting suggestion and question below.
>
> > 1. On the contrary, if a party has a constraint or minimal requirement on model performance instead of monetary budget, is your proposed optimisation framework applicable in this case? More concretely, if a party only requires the model to be as good as 90% of the best performing model in order to obtain some discounted payoffs, is your method applicable in this situation?
>
> We thank you for suggesting this important extension. We think that this can be incorporated by changing the budget constraint to a model performance constraint: The model performance is less than or equal to the minimum desirable performance.
>
> The reason for the "less than or equal to" (instead of larger than or equal to) is due to the budget efficiency property: A party expending as much of the budget as possible means it is getting as much performance (and additional payoff) as possible. Therefore, we need to constrain the model performance of a party to be less than or equal to its minimum desirable performance.
>
> Since the model performance of a party is still a linear function of $\lambda$, the resulting optimization problem is a linear program that can be solved efficiently.
> We will include this discussion in our revised paper.
>
> > 2. Another question: is the monotonicity assumption of s needed in your formulation? It seems to be needed to ensure joining the cooperation will be better than not joining, but in terms of Shapley values computation there is no such restriction needed when defining a value function? Can you comment a bit on this please?
>
> You are very observant in catching this subtle assumption. It is correct that the computation of the Shapley value does not require the monotonicity assumption. We only use the monotonicity assumption in the proof of the Budget efficiency in Appendix G.
>
> ---
>
> We would like to discuss your other comment below.
>
> > The introduction of payoff flow and conditional SVs weren't apparent to readers until getting to Sec.4, and it seemed to complicate the payoff allocation scheme. It was only made clear later in section 4 that we can handle the budget constraints by modifying the value functions of all parties in the Conditional SV. Maybe this intuition should be made a bit earlier to make it easier to follow.
>
> You have correctly understood the objective of the introduction of payoff flow and conditional SVs in our paper. As you may have noticed, we have briefly discussed the purpose of introducing payoff flows in the introduction (lines 71-72) and right before introducing payoff flows (lines 189-190). We will include the intuition behind payoff flows in our revised paper to make it more apparent to readers before getting to Section 4.
>
> ---
>
> We hope that the above clarifications will answer your insightful suggestion and comments.

---

### Official Review · Reviewer_iwL1 · 2022-07-11

**Rating:** 4
**Confidence:** 4
**Soundness:** 2 fair
**Presentation:** 2 fair
**Contribution:** 2 fair

**Summary:**

This paper considered the setting where there exists a tradeoff between payoff and model rewards in collaborative machine learning. Inspired by the observation that the difference between the contributions of parties should take into consideration during reward allocation, they proposed a new solution concept \emph{conditional Shapley value}, and proved their payoff function is unique for reasonable properties. Additionally, they decide their allocation based on the payoff function, and consider the settings that the parties have budget constraints, or without budget constraints, and empirically demonstrate the performance of their allocation scheme.

**Questions:**

1. Is there any rationale/literature on whether the party with more contributions should receive higher payoff rewards? From the reviewer's perspective, it's not a standard Fairness notion to assume this.

**Limitations:**

See weakness section.


**Strengths And Weaknesses:**

This paper has many strengths:
1. They consider a more general setting when the given model should of different payments based on the parties' contribution.
2. Their characterization of the desirable properties for a payoff reward vector produces a provable \emph{unique} payoff reward vector.
3. Their payoff allocation also satisfies replication-robustness that is studied in the previous literature.
4. The paper includes experiments on MNIST, CIFAR-10, and IMDB datasets.

Weaknesses:
1. The paper doesn’t compare against the existing benchmark either in theoretical statements or empirically.
2. No contribution section in the introduction. ([line 65-90] has an overview of the sections, but doesn't include the theoretical results of this paper.
3. The descriptions of the payoff functions are hard to follow.
4. No running time analysis.

=====================================================================
After review rebuttal period, the reviewer would like to add more weakness as follows:

1. Results for similar settings, though not explicitly stated as "Fair Allocation", has been published in 2019, see [1].

2. Many of the concepts in this paper are stated without appropriate references, for example, "Individual Rationality", "Marginal contribution consistency", "balance", "linear Value", "dummy party" etc. These concepts are widely used metrics for mechanism design and beyond.
3. The conditional Shapley value seems to be similar to Shapley value, as in stated in Remark 3.5. Additionally, for the properties introduced in this paper, i.e., balance, linear Value, dummy party, similar properties are already been proved as in [2] for Shapley value, which makes the theoretical contributions for providing such metric incremental.

3. This paper doesn't have comprehensive references, at least existing literature on collaborative learning/ allocations rules. From the reviewer's perspective, while there are total 17 references, many of them are not directly related to collaborative learning. Additionally, the current references are informal, i.e., without the conference/ journal name, e,g. [15], [16], [8] in the paper.

The reviewer is from a more theoretical background, so my evaluations are more on the theoretical contributions of this paper, i.e., the novelty of the proof for theorem statements, and how the results are different from existing metrics. From the reviewer's perspective, the writing of this paper itself (references, connection to existing results ) needs a major revision.

[1] Ghorbani, A., & Zou, J. (2019, May). Data Shapley: Equitable valuation of data for machine learning. In International Conference on Machine Learning (pp. 2242-2251). PMLR.

[2] Agarwal, A., Dahleh, M., & Sarkar, T. (2019, June). A marketplace for data: An algorithmic solution. In Proceedings of the 2019 ACM Conference on Economics and Computation (pp. 701-726).

---

> ### Author Response · Authors · 2022-08-02
> **Response to Reviewer iwL1 (part 1)**
>
> We would like to thank you for recognizing many strengths of our work and the helpful suggestions.
>
> ---
>
> We would like to address your question on the standard fairness notion: "the party with more contributions should receive higher payoff rewards" below.
>
> > Is there any rationale/literature on whether the party with more contributions should receive higher payoff rewards? From the reviewer's perspective, it's not a standard Fairness notion to assume this.
>
> We think it is reasonable that the party with more contributions should receive higher payoff rewards. For example, the author that contributes the most to a paper receives the highest reward, e.g., by being the lead author of the paper (most recognized author). Investors that contribute more to a company (e.g., by investing more money or spending more time working with the company) receive a larger number of shares in the company. More concretely, two people A and B buy a piece of land together at the price of 500,000 dollars for which A pays 100,000 dollars and B pays 400,000 dollars. Later, the piece of land is sold for 1 million dollars. It will be unfair if A and B receive the same payoff of 500,000 dollars each. In fact, to be fair, it should be the case that A receives 200,000 dollars and B receives 800,000 dollars from the sale of the land because the amount of money that A contributes to the purchase of the land is only 1/4 that B contributes.
>
> Many solution concepts to distributing payoff fairly in the collaborative ML reflect this intuition. For example, the Shapley value, the Banzhaf value, and leave-one-out of a party are all linear combinations of marginal contributions of the party to coalitions with non-negative weights. It means that an increase/decrease in the contribution of a party leads to an increase/decrease in the payoff reward and parties with the same contributions to all coalitions receive the same payoff reward.

---

> > ### Author Response · Authors · 2022-08-02
> > **Response to Reviewer iwL1 (part 2)**
> >
> > Below are the clarifications on some contents of our paper that you may have missed in our paper.
> >
> > > 1. The paper doesn’t compare against the existing benchmark either in theoretical statements or empirically.
> >
> > We would like to highlight that our work is the first to address the trade-off between payoff and model rewards. Hence, in our general problem settings, there aren't any existing baseline methods to compare with.
> >
> > Nonetheless, we compared against two existing solutions in special cases of our problem settings.
> > * Theoretically,
> >   * in lines 58-61, existing solutions [12,14] can only work in special cases of our problem settings (either using monetary payoff or the model reward alone), while our proposed scheme can allocate both monetary payoff and model reward at the same time (by trading off between them).
> >   * in lines 61-64, even in the special cases of our problem settings where [12,14] are applicable, our solution is different from them because ours satisfy the linearity property while they do not.
> >   * in lines 153-157, existing works of [12,14] directly scale the Shapley value to correct the problem of inefficiency, while our scheme satisfies the balance property without any scaling.
> > * Empirically, we have performed experiments in the special cases where the works of [12,14] are applicable (using only the monetary payoff or only the model reward). In lines 362-364, we computed the solution of [12] and showed that it is different from ours, which agrees with the theory since the solution of [12] does not satisfy the linearity property while ours does. In lines 371-374, we also commented that our solution cannot be achieved by varying the parameters of the $\rho$-Shapley value in [14] because of the linearity property. The same observations are stated in 2 experiments in Appendices H.1 and H.2.
> >
> >
> > > 2. No contribution section in the introduction. ([line 65-90] has an overview of the sections, but doesn't include the theoretical results of this paper.
> >
> > You may have missed the summary of our main contribution which is described in lines 58-64 (in the Introduction section). It is about resolving 2 important issues of existing works that are motivated in lines 48-57: trade-off between payoff and model rewards and the uniqueness of the schemes.
> >
> > Lines 65-90 are not only an overview of the sections, but they also elaborate our contributions in detail, including the theoretical contributions: (lines 67-76) uniqueness of the schemes without budget constraints, the proposed fair payoffs, the proposed conditional Shapley value; (lines 81-84) the uniqueness of the adjustment of the allocation scheme under budget constraints.
> >
> > Would you please let us know the theoretical results that you are looking for in the Introduction section? We will incorporate them accordingly in our revised paper.
> >
> >
> > > 3. The descriptions of the payoff functions are hard to follow.
> >
> > To facilitate the understanding of the payoff flow, we borrow the business concepts of cost and revenue in lines 193-196. We also visualize the payoff flows in Figure 1b. We will include a more detailed explanation in our revised paper.
> >
> > > 4. No running time analysis.
> >
> > We have discussed the computational aspect of our approach in Remark 3.5: It is the same as the Shapley value (exponential in the number of parties), and existing approximation methods of the Shapley value [2,4,11] can be used. We will elaborate it in more detail in our revised paper.
> >
> > ---
> >
> > We sincerely hope that the above clarifications will improve your opinion of our work.

---

> > > ### Comment · Reviewer_iwL1 · 2022-08-06
> > > **Thanks for the response.**
> > >
> > > Thanks for the review response. The reviewer changed the score accordingly.

---

> > > > ### Author Response · Authors · 2022-08-06
> > > > **Seeking further clarification on the decision**
> > > >
> > > > Thank you for agreeing with our response. Please let us know if you have any remaining concern underlying your current decision. We will do our best to address it (if any) with the remaining time.

---

> > > ### Comment · Reviewer_iwL1 · 2022-08-07
> > > **Other Concerns of this paper**
> > >
> > > Another few issues I think is not minor is that:
> > > - The paper doesn't have enough literature reviews on existing literature on collaborative learning/ fair allocations rules. From the reviewer's perspective, while there are total 17 references, many of them are not directly related to collaborative learning.
> > >
> > > - Also, when the reviewer looks into the detail of the references, many of the paper references are informal, i.e.,  without the conference/ journal name, e,g. [15], [16], [8].
> > >
> > > - Regarding the benchmark "replication robustness": This benchmark hasn't been established for top ML conferences like ICML, NeurIPS, etc, so the reviewer doesn't see there is enough importance/research attention of this benchmark. If there are more existing literature to justify the importance of this benchmark, please include it in the references.
> > >
> > > - Regarding the importance of the properties, i.e., Balance, Linear Value, Dummy party, there should be support on the generality of these metrics, as well as the papers that regard them as important properties. Also, when the reviewer digs into the context of [1], it seems to me **the similar properties are already been proved as in [1], i.e., section 3.3.1 Shapley Fairness**, where:
> > > -- The "balance" property corresponds to prop 3.3 (1) in [1]
> > > -- The "dummy party" property corresponds to prop 3.3 (3) in [1]
> > > -- The "Linear Value" property corresponds to prop 3.3 (4) in [1]
> > >
> > > Additionally, the proof for uniqueness of the benchmark doesn't establish enough technical novelty in theory.
> > > Hence, it seems to the reviewer that the theoretical contributions of this paper is incremental.
> > >
> > > -Additionally, for NeurIPS rebuttal this year, the author can upload a **revised** version of the paper. Instead of saying "I will do it in the later version", please plug in the constructive comments by all the reviewers into the version, since we don't know whether these changes would actually happen.
> > >
> > > As a result, the reviewer would still remain the current score of this paper.
> > >
> > > [1]: Agarwal, A., Dahleh, M., & Sarkar, T. (2019, June). A marketplace for data: An algorithmic solution. In Proceedings of the 2019 ACM Conference on Economics and Computation (pp. 701-726).

---

> > > > ### Author Response · Authors · 2022-08-09
> > > > **Clarifications on the Other Concerns (part 1)**
> > > >
> > > > We would like to thank the reviewer for the response. Since the reviewer has not repeated (nor asked for further clarification on) any existing weaknesses in the initial reviews, we assume that they are more or less addressed. So, we focus on addressing the new issues in the latest post (on 7 Aug) and some newly added comments after the initial review (on 9 Aug).
> > > >
> > > > In our response below, the **[latest comment]** refers to the post on 7 Aug and the **[added review]** refers to the additional review added to the initial review (i.e., the part after the double dashed line in the initial review) on 9 Aug.
> > > >
> > > > Also, the citation numbers in the latest comment and the added review are inconsistent. Since these works are cited in our paper, we use the citation numbers from the references in our paper (e.g., **[1] for the work on the marketplace for data** and **[4] for the Data Shapley work**).
> > > >
> > > > > [latest comment] Regarding the importance of the properties, i.e., Balance, Linear Value, Dummy party, there should be support on the generality of these metrics, as well as the papers that regard them as important properties. Also, when the reviewer digs into the context of [1], it seems to me the similar properties **are already been proved as in [1], i.e., section 3.3.1 Shapley Fairness**, where: -- The "balance" property corresponds to prop 3.3 (1) in [1] -- The "dummy party" property corresponds to prop 3.3 (3) in [1] -- The "Linear Value" property corresponds to prop 3.3 (4) in [1]
> > > >
> > > > > [added review] Many of the concepts in this paper are stated without appropriate references, for example, "Individual Rationality", "Marginal contribution consistency", "balance", "linear Value", "dummy party" etc. These concepts are widely used metrics for mechanism design and beyond.
> > > >
> > > > > [added review] Results for similar settings, though not explicitly stated as "Fair Allocation", has been published in 2019, see [1] ([4] in our paper)
> > > >
> > > > Firstly, since the properties in Section 3.3.1 of [1] are the Shapley value (SV)'s properties from the seminal paper [13] (i.e., taken from the paragraph after Property 3.3 in [1]), let us just refer to these properties as the SV properties in short. SV properties have inspired many works including those in the collaborative ML and data valuation literature (including [1] and [4]). As we are also inspired by the SV properties, we cited the work [13] and **mentioned these 4 properties in lines 108-109** in our paper (and the full description of these properties are in **Appendix A**). Besides, we have analyzed the relationship between our solution and the Shapley value in Equation 6 (i.e., **the decomposition of the Shapley value**) and explained why SV **cannot be directly used** in our settings in lines 112-118 and Remark 3.3.
> > > >
> > > > Secondly, at a high level, it is **not obvious** how  Shapley value can be used to answer the following questions that are addressed in our paper:
> > > > * Can we use the Shapley value to distribute rewards to other parties if there is no reward to begin with? One may think that each party can pay an amount of reward to generate the common reward to be distributed, but the amount of payment is debatable (in fact, the same significant question is raised by [14] but without an answer).
> > > > * Can we use the Shapley value to trade off between payoff and model rewards when there are budget constraints?
> > > >
> > > > We want to highlight that **neither the solution in [1] nor that in [4] resolves the above questions**.
> > > >
> > > >   Regarding the problem setting in [4], as stated in our introduction, **[4] belongs to the group of existing works using data valuation methods (lines 33-39)**. Our work belongs to the other group (based on our proposed classification of existing works in the introduction), as stated in lines 40-43. Hence, the problem settings are **clearly different**.
> > > >
> > > >   Thirdly, the definition of some properties in our works are **different** from the SV properties. For example,
> > > > * Our balance property states that the sum of payoff rewards is $0$, while the efficiency property of SV states that the sum of payoff rewards is $v(\mathcal{N})$. **Mathematically**, $v(\mathcal{N}) > 0$ in our setting (line 98). **Intuitively**, $v(\mathcal{N})$ is the value of the grand coalition which should not be interpreted as a constant $0$. As the allocation scheme and its uniqueness depend on the whole set of fairness properties, changing some properties will lead to a different solution. Our proposed dummy party property is also different from that of SV for a similar reason.
> > > >
> > > > * Our proposed linear value property implies that $p_i(v)$ is a linear function **of SV**, while the linearity property of SV implies that SV is linear **in the value function**. Hence, they are different.
> > > >
> > > > Regarding the marginal contribution consistency, we have designed it specifically to handle the budget constraints. Hence, to the best of our knowledge, **it has not been discussed before in the literature**.

---

> > > > > ### Author Response · Authors · 2022-08-09
> > > > > **Clarifications on the Other Concerns (part 2)**
> > > > >
> > > > >   > [latest comment] Regarding the benchmark "replication robustness": This benchmark hasn't been established for top ML conferences like ICML, NeurIPS, etc, so the reviewer doesn't see there is enough importance/research attention of this benchmark. If there are more existing literature to justify the importance of this benchmark, please include it in the references.
> > > > >
> > > > >   We have intended the replication-robustness result to be an auxiliary result for interested readers; it is only mentioned in 2 lines 182-183 in our main paper. Do also note that replication robustness is not mentioned as our main contribution in the introduction.
> > > > >
> > > > >   On the other hand, we think that replication robustness is a well-motivated problem in existing works [1,5], but due to the page limit and the focus of our paper, we refer readers to [1,5] for a detailed discussion.
> > > > >
> > > > >   > The paper doesn't have enough literature reviews on existing literature on collaborative learning/ fair allocations rules. From the reviewer's perspective, while there are total 17 references, many of them are not directly related to collaborative learning.
> > > > >
> > > > >   We structure our literature review by dividing existing works into 2 groups (lines 28-32). The first group contains many existing works on data valuation methods (lines 33-39). We position our work in the second group where, to the best of our knowledge, there are 2 existing works that are most relevant to ours (lines 40-57).
> > > > >
> > > > >   > Also, when the reviewer looks into the detail of the references, many of the paper references are informal, i.e., without the conference/ journal name, e,g. [15], [16], [8].
> > > > >
> > > > >   We would like to thank the reviewer for spotting the errors in the references. They are indeed due to our careless typoes in our BibTeX file. We will correct them accordingly.
> > > > >
> > > > >   > Additionally, for NeurIPS rebuttal this year, the author can upload a revised version of the paper. Instead of saying "I will do it in the later version", please plug in the constructive comments by all the reviewers into the version, since we don't know whether these changes would actually happen.
> > > > >
> > > > >   To address the reviewers' comments/concerns, we have elaborated the answers in our response to indicate that they can be resolved with confidence, as acknowledged by the other reviewers.
> > > > >
> > > > >   As there are only less than 3 days before the discussion deadline, we kindly seek your understanding for not acceding to your request to "plug in the constructive comments by all the reviewers into a version". This is because our current paper fits the submission page limit exactly, so incorporating new content at this time requires, for example, moving some parts to the appendix and some rearranging in this time-critical period. We understand that there would be an additional page given to accepted papers to incorporate the new content, which significantly eases this task. Since the reviewers' comments are constructive, we like to reassure the reviewer that they will be incorporated them into our revised paper; we see no reason for not doing so.
> > > > >
> > > > > > [added review] The conditional Shapley value seems to be similar to Shapley value, as in stated in Remark 3.5. Additionally, for the properties introduced in this paper, i.e., balance, linear Value, dummy party, similar properties are already been proved as in [2] ([1] in our work) for Shapley value, which makes the theoretical contributions for providing such metric incremental.
> > > > >
> > > > > > [latest comment] Additionally, the proof for uniqueness of the benchmark doesn't establish enough technical novelty in theory. Hence, it seems to the reviewer that the theoretical contributions of this paper is incremental.
> > > > >
> > > > > Firstly, we would like to highlight that we follow an axiomatic approach (as noted by Reviewer Bdvn). Hence, there is no need to prove the fairness properties. They are viewed as axioms which are assumed to be true. On the other hand, we justify their relevance to our setting in lines 151-152 (balance), lines 160-165 (linear value), and lines 166-170 (dummy party).
> > > > >
> > > > > Overall, our theoretical contribution is about resolving the **2 important issues** discussed in lines 48-57, which have **not been addressed before** in the literature. We also highlight that the theoretical contribution is not only about the proofs but also about choosing a set of properties that (i) allows the proof to go through and (ii) unifies both cases: with and without budget constraints (i.e., trading off between payoff and model rewards). In other words, we think laying the groundwork for the proof and the unification to work naturally is another important technical contribution that differentiates our work from the existing literature.
> > > > >
> > > > > We sincerely hope that the above clarifications will help improve your opinion of our paper.

---

### Official Review · Reviewer_Bdvn · 2022-07-11

**Rating:** 6
**Confidence:** 4
**Soundness:** 3 good
**Presentation:** 2 fair
**Contribution:** 2 fair

**Summary:**

The paper considers collaborative machine learners where multiple vendors combine their data so as the generate more accurate models than either of them can achieve on their own. The authors focus on tradeoffs between payoff and model rewards, including for budget restricted vendors. The method proposed can be derived from two methods,  based on desirable properties of the parties’ payoffs or based on the underlying payoff flows from one party to another (the second allows handling constraints on the budgets of vendors). The authors propose desiderata for buiding a fair adjustment of the payoff flows that trades off between the model reward’s performance and the payoff reward. The authors also support their claims on several collaborative ML scenarios.

Thansk for the detailed author comments.

**Questions:**

Are there any guarantees on the shared data? Can one vendor "poison" the data given to others so as to achieve its goals? See discussion in: Meir, Reshef, Ariel D. Procaccia, and Jeffrey S. Rosenschein. "Algorithms for strategyproof classification." Artificial Intelligence 186 (2012): 123-156.

Further, can other foundations of mechanism design (such as auctions or the Vickery-Clarke-Groves mechanism) be used here for similar applications (e.g. auction off the data by a vendor, rather than redistributing payments).

Could there be other constraints (e.g. computational ones) that be tackled here, rather than the budget constraints.

Finally, you show that your solutions are "sensible" and result with reasonable outcomes. What kind of additional metrics could quantify that? What could one show in future work to demonstrate an even "more fair" outcome?

**Limitations:**

This builds heavily on earlier work and foundations from cooperative game theory. The empirical analysis is illuminating but limited (few domains, few metrics). The authors should do more to justify their choices of desiderata (and surey relevant game theory work on how different desiderata yield different solutions).

While the question of whether budget constraints are reasonable ones for ML vendors is subjective, the authors focus on this one constraint.

**Strengths And Weaknesses:**

I find the topic of the paper very interesting. There have recently been multiple works on mechanisms for allowing vendors to share their data, focusing on how to incentivize them to do so by fairly sharing the proceeds of the resulting model (or by yielding smaller models where the quality of a model a party receives depends on its contribution). The key strength of the paper is by combining key methodology from cooperative game theory, such as the Shapley value and the desiderata yielding this value, and applying them to this domain.

However, given that there are already multiple papers on this topic, I believe the bar for additional work here is higher. The paper has some weaknesses. First, while taking an axiomatic (desiderata) approach, I'm not sure I fully by into the proposed desiderata here. Specifically, for the Shapley value which is a common theme here, one of the axioms is linearity across games. Other powre indices are triggered by alternative approaches. See, for e.g. (and I'd encourage the authors to discuss these, in more depth):

Dubey, Pradeep. "On the uniqueness of the Shapley value." International Journal of Game Theory 4.3 (1975): 131-139.
Dubey, Pradeep, and Lloyd S. Shapley. "Mathematical properties of the Banzhaf power index." Mathematics of Operations Research 4.2 (1979): 99-131.Feltkamp, Vincent. "Alternative axiomatic characterizations of the Shapley and Banzhaf values." International Journal of Game Theory 24.2 (1995): 179-186.
Monderer, Dov, and Dov Samet. "Variations on the Shapley value." Handbook of game theory with economic applications 3 (2002): 2055-2076.
(and see the very related discussion in Torra, Vicenç, and Yasuo Narukawa. Modeling decisions: information fusion and aggregation operators. Springer Science & Business Media, 2007.)

Further, regarding writing, I feel like a more detailed introduction in cooperative game theory is warranted. You build on these foundations, but the discussion is short. For example, what are the key differences between such power indices and other known methods such as the Core or Nucleolus? Why are these less applicable here?

Finally, I suggest addressing computational constraints. What is the exact time (and memory) complexity of the proposed approaches? Are you computing Shapley values exacly, or approximating? Can alternative computational mechanisms be used?

---

> ### Author Response · Authors · 2022-08-02
> **Response to Reviewer Bdvn (part 1)**
>
> We would like to thank you for your detailed feedback and references which we will incorporate into the revised paper. We are glad that you find the topic of our work interesting.
>
> ---
>
> We would like to address your questions below.
>
> > Are there any guarantees on the shared data? Can one vendor "poison" the data given to others so as to achieve its goals? See discussion in: Meir, Reshef, Ariel D. Procaccia, and Jeffrey S. Rosenschein. "Algorithms for strategyproof classification." Artificial Intelligence 186 (2012): 123-156.
>
> Our main goal in this work is to design fair allocation schemes of both model and payoff rewards. Hence, we assume that parties do not "poison" their training datasets. We will clarify this assumption in our revised paper.
>
> On the other hand, we consider the replication attack in lines 182-183 in our paper. This is a common type of attack that is considered in existing collaborative ML works [5,12].
>
> We think that mechanisms to prevent/discourage the poisoning attack while still maintaining fairness is an important, yet challenging, problem that can be an interesting next step for us.
>
> > Further, can other foundations of mechanism design (such as auctions or the Vickery-Clarke-Groves mechanism) be used here for similar applications (e.g. auction off the data by a vendor, rather than redistributing payments).
>
> The reason that existing mechanism designs such as auctions and the Vickery-Clarke-Groves mechanism are not directly applicable to the collaborative ML setting is that the training data are shared among all parties, unlike a painting at an auction that is only sold to the winner. In particular, commodities in auctions and the Vickery-Clarke-Groves mechanism often cannot be shared/used by multiple parties, while the dataset of a party can be used in $n$ models (for $n$ parties), as mentioned in Remark 3.3 in our paper. This property of data is also known as free replicability in [1,14] which makes data different from the normal commodities.
>
> We also like to highlight that the traditional Shapley value cannot be directly used in our setting because there is no external source of reward (lines 112-118). However, we find it easier to rely on the Shapley value's properties (instead of auctions) to design different allocation schemes by proposing the conditional Shapley value (without budget constraints) and formulating a linear programming problem (with budget constraints) while still retaining certain desirable fairness properties and the uniqueness of the schemes.
>
> > Could there be other constraints (e.g. computational ones) that be tackled here, rather than the budget constraints.
>
> While our work does not address the computational cost in training the aggregated dataset, one naive solution we think of is to convert the computational cost to the monetary cost so that it can be subtracted/added from the payoff rewards. For example, a party with a negative payoff has its payment deducted by an equivalent amount of their contributed computation, or a party with a positive payoff has its payoff increased by an equivalent amount of their contributed computation.
>
> However, to constrain the computational cost, we need to investigate its implication more deeply. For example, if the training cannot be completed (due to computational constraints), then the value of a dataset cannot be simply defined as the (optimal) model performance as in our current work.
>
> > Finally, you show that your solutions are "sensible" and result with reasonable outcomes. What kind of additional metrics could quantify that? What could one show in future work to demonstrate an even "more fair" outcome?
>
> We show that our proposed schemes are fair from 2 aspects: (i) the desirable properties (e.g., balanced, linear value, dummy party) that uniquely define the schemes and their derived properties (e.g., omnipotent party); (ii) some empirical observations from the experiments (in the 4 bullet points that summarize common empirical observations in lines 349-355). Notably, compared with existing works [12,14], our proposed schemes satisfy the linearity property (a fairness property of the Shapley value) while the solutions of [12,14] do not.
>
> Due to the uniqueness of our schemes, it is not possible to design a different allocation scheme that satisfies the same set of desirable properties for fairness.
> Hence, for future work, to argue that it is "more fair", it needs to be based on a different set of properties for fairness. For example, as you have suggested, one may be inspired by other solution concepts of coalitional games, e.g., the core which requires the property of coalitional rationality, or the nucleolus which minimizes the excess of coalitions. Depending on the scenario, one may prefer a set of properties over another.

---

> > ### Author Response · Authors · 2022-08-02
> > **Response to Reviewer Bdvn (part 2)**
> >
> > We would like to address your other concerns below.
> >
> >
> > > First, while taking an axiomatic (desiderata) approach, I'm not sure I fully by into the proposed desiderata here. Specifically, for the Shapley value which is a common theme here, one of the axioms is linearity across games. Other powre indices are triggered by alternative approaches.
> >
> > We would like to clarify that the proposed desiderata have led to several reasonable and interesting properties in both theory (e.g., the omnipotent party property, being interpreted as the conditional Shapley value which is a variant of the famous Shapley value) and experiments (see the 4 bullet points in lines 349-355).
> >
> > Let us consider a simple example to justify the linearity property: When the exchange rate (defined in line 105) increases by 2 times, we naturally expect that the payoff of each party has its absolute value increase by 2 times. The analogy is that when the price of an item increases by 2 times, both the seller's revenue and the buyer's payment increase by 2 times. Hence, existing solutions in [12,14] that violate the linearity do not afford such an intuitive implication in this scenario. Besides, the motivation of the linearity property follows from that of the Shapley value.
> >
> > > Further, regarding writing, I feel like a more detailed introduction in cooperative game theory is warranted. You build on these foundations, but the discussion is short. For example, what are the key differences between such power indices and other known methods such as the Core or Nucleolus? Why are these less applicable here?
> >
> > We do not claim that the Shapley value is more applicable than the core and the nucleolus in collaborative ML.
> > We simply choose the properties of the Shapley value because they allow us to address the 2 important issues mentioned in lines 40-57 (trading off between payoff and model rewards, and the uniqueness of the schemes with budget constraints) which, to the best of our knowledge, have not been addressed by any works in the collaborative ML and the cooperative game theory literature. We think it is a significant contribution to be the first work that addresses these issues.
> >
> > Having said that, one may choose to use the core and the nucleolus solution concepts. However, just like in our work, they will need to address several challenges: the difference in the problem settings of collaborative ML vs. coalitional games, and maintaining the fairness of the allocation scheme under budget constraints. Personally, we think that using these concepts to address our problem can pose an even bigger challenge. Regarding the core, it is a set solution concept that includes many cases such as the core containing multiple payoff vectors and the core being empty, which may make it difficult to come up with a fair adjustment under budget constraints. Regarding the nucleolus, while it is a single-valued solution concept like the Shapley value, it involves minimizing the excess of coalitions (an optimization problem), unlike the Shapley value which has a closed-form expression. This may make it challenging to efficiently address the budget constraints. We will include further comparison and this discussion in our revised paper.
> >
> > Besides, our work is positioned in the collaborative machine learning (ML) literature, so given such a research context, the general perception is that the Shapley value is a popular solution concept (e.g., in existing works [1,3,4,5,7,12,14,15] that are briefly reviewed in lines 37-45). In fact, we could hardly find any comparison between the Shapley value and the core and nucleolus in these existing works. Since we are pushing the frontier of this line of Shapley-value-based works by considering the trade-off between payoff and model rewards, our background section is mainly about the Shapley value (lines 108-118 and Appendix A).

---

> > > ### Author Response · Authors · 2022-08-02
> > > **Response to Reviewer Bdvn (part 3)**
> > >
> > > > Finally, I suggest addressing computational constraints. What is the exact time (and memory) complexity of the proposed approaches? Are you computing Shapley values exacly, or approximating? Can alternative computational mechanisms be used?
> > >
> > > We like to clarify that the computation of the Shapley value has been discussed in Remark 3.5: It is the same as that of the Shapley value which is exponential in the number of parties, and it is possible to apply existing approximation methods for the Shapley value [2,4,11]. Like existing works [12,14,15] where the focus is on fairness, we use the exact computation of the Shapley value in the experiments. The fair reward vector in Section 3.1 only depends on the Shapley value (Equation 2), so its time complexity is also the same as that of the Shapley value and approximation methods for the Shapley value can be used. Solving the linear program in Lemma 4.1 can be efficiently performed by any existing solver (e.g., the scipy solver in the attached source code). We will clarify the computational aspects in detail in our revised paper.
> > >
> > > > This builds heavily on earlier work and foundations from cooperative game theory. The empirical analysis is illuminating but limited (few domains, few metrics). The authors should do more to justify their choices of desiderata (and surey relevant game theory work on how different desiderata yield different solutions).
> > >
> > > We think that our work is significantly different from existing works on a similar problem such as [12,14], as elaborated in lines 40-57. Furthermore, to the best of our knowledge, the approach of trading off between payoff and model rewards has not been explored before in the cooperative game theory.
> > >
> > > We have performed a total of 3 experiments on the MNIST dataset (Section 5), the CIFAR-10 dataset (Appendix H.2), and the IMDB movie reviews dataset (Appendix H.3). In each experiment, we consider 3 different budget settings to illustrate the properties of the proposed schemes and compare our schemes with existing approaches. In each experiment, we also show the main components of our scheme: the payoff reward, the model reward, and the payoff flows between parties.
> > >
> > > Please let us know if there are additional metrics we can incorporate in our revised paper to enhance the understanding of the schemes.
> > >
> > > As one of our objectives is to identify a set of properties that uniquely define a scheme, altering any property can potentially break the desirable uniqueness characteristic (that is motivated in lines 52-57). Furthermore, justifying the necessity of the properties also implies that dropping one of these properties (e.g., non-symmetry, imbalance) is unsuitable in our problem setting. In particular, for the properties in our work that are different from that of the traditional Shapley value,
> > > * lines 151-152 justify the balance property;
> > > * lines 162-165 justify the linear value property;
> > > * lines 306-309 justify the feasibility and individual rationality property;
> > > * line 316 justifies the marginal contribution consistency.
> > >
> > > > While the question of whether budget constraints are reasonable ones for ML vendors is subjective, the authors focus on this one constraint.
> > >
> > > The problem of budget limitations is also highlighted in other existing collaborative ML works such as [12,14]. In particular, we motivate the need of considering budget constraints by highlighting the disadvantages of these two works in lines 43-57 in the Introduction section.
> > >
> > > ---
> > >
> > > We sincerely hope that the clarifications above will improve your opinion of our work.

---

### Official Review · Reviewer_GnPz · 2022-08-25

**Rating:** 6
**Confidence:** 4
**Soundness:** 3 good
**Presentation:** 3 good
**Contribution:** 3 good

**Summary:**

The paper studies the problem of fair payoff allocation of compensation/cost among a group of collaborating parties which pool their data to jointly train a machine learning model. Specifically, the authors considered several fairness properties (including a linearity property which helps to uniquely determines an allocation scheme). The authors then presented a unique allocation scheme which satisfies these properties. From there, the authors considered and resolved a more general setting where parties may not have sufficient budget by balancing their budget and model performances.

**Questions:**

Can the authors comment on the payoff allocation schemes that do not satisfy the linearity property?

**Limitations:**

I do not observe any negative societal impact.

**Strengths And Weaknesses:**

Strengths:
- the collaborative machine learning setting is interesting and important.
- the considered fairness axioms are reasonable, for example, the budge balance that keeps the total payoff/cost to zero among the group of parties.
- the proposed payoff allocation scheme is unique and supported with theoretical fairness axioms

Weaknesses:
- there is limited empircal comparison and analysis with prior works.

Some minor comments:
- the uniqueness of the allocation scheme is largely a result from the linearity axiom, that draws a connection to the Shapley value. I can see that the advantage of imposing the linearity is that it utilises the fairness of the Shapley value (a classic solution concept), which is a reasonable constraint for finding a unique solution. However, I do not see why allocation schemes that do not exhibit the linearity dependency on the Shapley value are inferior in the particular setting. This may be beyond the scope of the current paper, but a more comprehensive analysis on the class of payoff allocation schemes (without the linearity property) may be explored further in future works.  I also acknowledge the motivation and overall contributions of the present paper.
- As far as I know, there are quite a few papers on payoff allocation with external funds, but only a few on fully collaborative settings. I find this topic to be interesting for the community. I would not argue for acceptance weighing the strengths/weaknesses, but I am slightly positive hence a weak accept.

---

### Author Response · Authors · 2022-08-09
**Thank All Reviewers and The Area Chairs**

We would like to thank all reviewers and the area chairs for your time and effort in reading our paper, providing us with valuable feedback, and going through our detailed responses. We truly appreciate your suggestions and constructive feedback which we will carefully incorporate into our revised paper.

---

### Meta-Review · Area_Chair_jH39 · 2022-08-28

**Recommendation:** Accept
**Confidence:** Less certain

**Metareview:**

This paper lies in the borderline of acceptance, even with the help of an additional emergency expert reviewer (Reviewer GnPz) after the discussion phase:

On the positive side, the reviewers found the collaborative machine learning payoff problem studied in the paper interesting and relevant, and the fairness axioms reasonable, and the uniqueness proof of allocation schemes meaningful.

On the negative side, the reviewers found the contributions of this paper not groundbreaking. They still have the following concerns:
- The work is not placed well enough with existing work (comparison with existing works on fairness and Shapley value raised by the reviewers, and the references Reviewer Bdvn gave)
- The empirical comparison between prior works is limited (Reviewer GnPz)
- The time complexity for computing the (conditional) Shapley value is exponential, so it's not a metric of practical use. Moreover, variants of Shapley value have been already studied in the literature, see (Xu et al, Gradient driven rewards to guarantee fairness in collaborative machine learning.) (Reviewer iwL1)

**Award:**

No

---

### Decision · Program_Chairs · 2022-09-14

Accept